# Self-formation of concentric zones of telencephalic and ocular tissues and directional retinal ganglion cell axons

Wei Liu[1,2,3]*[†], Rupendra Shrestha[1,2,3], Albert Lowe[1,2], Xusheng Zhang[2], Ludovic Spaeth[4]

[1]Department of Ophthalmology and Visual Sciences, Albert Einstein College of Medicine, Bronx, United States; [2]Department of Genetics, Albert Einstein College of Medicine, Bronx, United States; [3]The Ruth L. and David S. Gottesman Institute for Stem Cell Biology and Regenerative Medicine, Albert Einstein College of Medicine, Bronx, United States; [4]Dominick P Purpura Department of Neuroscience, Albert Einstein College of Medicine, Bronx, United States

**\*For correspondence:**
wei.liu@einsteinmed.edu

[†]Lead contact

**Abstract** The telencephalon and eye in mammals are originated from adjacent fields at the anterior neural plate. Morphogenesis of these fields generates telencephalon, optic-stalk, optic-disc, and neuroretina along a spatial axis. How these telencephalic and ocular tissues are specified coordinately to ensure directional retinal ganglion cell (RGC) axon growth is unclear. Here, we report self-formation of human telencephalon-eye organoids comprising concentric zones of telencephalic, optic-stalk, optic-disc, and neuroretinal tissues along the center-periphery axis. Initially-differentiated RGCs grew axons towards and then along a path defined by adjacent PAX2+ VSX2+ optic-disc cells. Single-cell RNA sequencing of these organoids not only confirmed telencephalic and ocular identities but also identified expression signatures of early optic-disc, optic-stalk, and RGCs. These signatures were similar to those in human fetal retinas. Optic-disc cells in these organoids differentially expressed *FGF8* and *FGF9*; FGFR inhibitions drastically decreased early RGC differentiation and directional axon growth. Through the RGC-specific cell-surface marker CNTN2 identified here, electrophysiologically excitable RGCs were isolated under a native condition. Our findings provide insight into the coordinated specification of early telencephalic and ocular tissues in humans and establish resources for studying RGC-related diseases such as glaucoma.

## eLife assessment

In this **important** study, the authors present a human telencephalon-eye organoid model that exhibits remarkable pathfinding and growth of retinal ganglion cell (RGC) axons. The identification of cell-surface markers for RGCs could have value for understanding the molecular mechanisms involved in RGC axon development and regeneration. The strength of evidence is **compelling** for future studies to investigate RGC neurite outgrowth and brain-eye connectivity in humans.

## Introduction

Our understanding of eye and brain development in humans is mostly deduced from animal studies. In mice, fate mapping of the anterior neural plate reveals that the eye field is in rostral regions surrounded anteriorly and laterally by the telencephalic field and caudally and medially by the hypothalamic field, indicating the proximity of their embryonic origins (*Inoue et al., 2000*). Subsequent evagination of the eye field generates bilateral optic vesicles and optic stalks, and the optic stalk connects the optic

vesicle to the forebrain. The optic vesicle then invaginates ventrally, resulting in the formation of the double-layered optic cup in which the inner and outer layers develop into the neuroretina and retinal pigment epithelium (RPE), respectively. The posterior pole of the optic cup forms the optic disc (also known as optic nerve head). In the central neuroretina close to the nascent optic disc, retinal ganglion cells (RGCs) start to appear as the optic fissure nearly closes. Early RGC axons find their path toward the optic disc and then enter the optic stalk to reach the brain. Concentrically organized growth-promoting and growth-inhibitory guidance cues around the optic disc regulate RGC axon growth and pathfinding through multiple mechanisms (*Erskine and Herrera, 2007*; *Erskine and Herrera, 2014*; *Oster et al., 2004*). Therefore, early eye morphogenesis leads to coordinated specification of the telencephalic, optic stalk, optic disc, and retinal tissues along a spatial axis.

Early telencephalic and eye development is marked and regulated by a group of tissue-specific transcription factors and signal transduction molecules (*Wilson and Rubenstein, 2000*). In mice, Foxg1 is specifically expressed in the presumptive telencephalon and is essential for the development of the cerebral hemispheres (*Xuan et al., 1995*). Pax6 is specifically expressed in the eye field and is essential for the development of multiple ocular tissues, such as the neuroretina, lens, and RPE (*Ashery-Padan et al., 2000*; *Bäumer et al., 2003*; *Marquardt et al., 2001*; *Zuber et al., 2003*). Vsx2 and Mitf are specifically expressed in the neuroretina and RPE, respectively, and are essential for retinal development (*Bharti et al., 2012*; *Horsford et al., 2005*; *Liu et al., 2010*; *Rowan et al., 2004*). Pax2 is expressed in the optic stalk, optic vesicles, central neuroretina, and optic disc; Pax2 is essential for optic stalk and nerve development (*Macdonald et al., 1997*; *Martinez-Morales et al., 2001*; *Schwarz et al., 2000*; *Torres et al., 1996*). Fgf8 is specifically expressed at the rostral forebrain at early stages, induces Foxg1 expression (*Shimamura and Rubenstein, 1997*), and regulates telencephalic patterning in a dose-dependent manner (*Storm et al., 2006*). Fgf8 also maintains Pax2 expression in the optic stalk (*Soukkarieh et al., 2007*), and Fgf8 and Fgf3 coordinate the initiation of retinal differentiation in chicks (*Martinez-Morales et al., 2005*; *Vogel-Höpker et al., 2000*). Despite these findings in vertebrates, it is unclear how telencephalic and ocular tissues are specified in humans.

Human organoids are transformative since they enable us to study the mechanisms of cell specification and differentiation directly in human tissues (*Lancaster and Knoblich, 2014*). Self-organized three-dimensional (3-D) retinal organoids are originally reported by Sasai's group and further improved in follow-up studies (*Cowan et al., 2020*; *Eiraku et al., 2011*; *Fligor et al., 2021*; *Kim et al., 2019*; *Lowe et al., 2016*; *Meyer et al., 2011*; *Nakano et al., 2012*; *Reichman et al., 2014*; *Zhong et al., 2014*). We and others have demonstrated that 3-D retinal organoids derived from human embryonic stem cells (hESCs) display a stratified structure containing all major types of retinal cells. Although RGCs are differentiated in 3-D retinal organoids, there is no proper RGC axon outgrowth toward the optic disc as seen in vivo since the optic disc-like tissue is missing in 3-D retinal organoids. When these organoids are dissociated into single cells or cut into pieces for adherent culture, RGCs generate neurites (*Fligor et al., 2018*; *Langer et al., 2018*; *Teotia et al., 2017*). A variety of brain organoids have been described (*Kadoshima et al., 2013*; *Lancaster et al., 2013*; *Mariani et al., 2012*; *Qian et al., 2017*; *Velasco et al., 2019*). Although rudimentary ocular tissues are occasionally found in some brain organoids (*Gabriel et al., 2021*), optic disc- and stalk-like tissues are absent. Collectively, RGC axon outgrowth and pathfinding directed by optic disc- and stalk-like tissues in organoids have not been reported.

Tissue patterning and coordinated specification are fundamental for body plan formation in vivo. Remarkably, concentric zones of trophectoderm, endoderm, mesoderm, and ectoderm (*Etoc et al., 2016*; *Minn et al., 2020*), neural plate and neural plate border (*Xue et al., 2018*), and ectodermal cells (*Hayashi et al., 2016*) are self-organized from single pluripotent stem cells, indicating patterning and coordinated specification of these tissues in vitro. Nevertheless, optic disc and stalk tissues are not reported in any of these structures.

We hypothesize that coordinated specification from the anterior ectodermal epithelium via morphogen gradients leads to self-organization of telencephalic and ocular tissues, including optic disc- and stalk-like tissues that provide guidance cues for RGC axon growth and pathfinding. In support of this hypothesis, we generated self-formed human telencephalon-eye organoids that comprise *concentric zones of anterior ectodermal progenitors* (CONCEPT), including FOXG1+ telencephalic, PAX2+ VSX2- optic stalk, PAX2+ VSX2+ optic disc, and VSX2+ neuroretinal cells along the center-periphery axis. We call this system as an organoid since it displays a spatially organized structure

with cell identities mimicking those tissues in vivo. In CONCEPT organoids, early differentiated RGCs grew their axons toward and then along a path defined by adjacent PAX2+ VSX2+ optic disc cells. Single-cell RNA sequencing of CONCEPT organoids not only confirmed telencephalic and ocular identities but also discovered expression signatures of cell clusters. Additionally, CONCEPT organoids and human fetal retinas had similar expression signatures. Furthermore, PAX2+ VSX2+ optic disc cells differentially expressed FGF8 and FGF9; inhibition of FGF signaling with an FGFR inhibitor during early RGC differentiation drastically decreased the number of RGCs somas and nearly ablated directional axon growth. Using the RGC-specific cell-surface marker CNTN2 identified in our single-cell RNA-sequencing, we developed a one-step method for isolating electrophysiologically-excitable RGCs under a native condition. Our studies provide deeper insight into the coordinated specification of telencephalic and ocular tissues in humans and establish resources for studying neurodegenerative diseases such as glaucoma.

## Results

### Generation of telencephalon-eye organoids that are composed of *concentric* zones of anterior *ectodermal progenitors* (CONCEPT)

The telencephalon and eye in mammals are originated from adjacent fields at the anterior neuroepithelium (*Inoue et al., 2000*). Morphogenesis of these embryonic fields leads to the formation of telencephalon, optic stalk, optic disc, and neuroretina along a spatial axis. Early differentiated retinal ganglion cells (RGCs) in the neuroretina grow axons toward the optic disc and then along the optic stalk to reach the brain. How these telencephalic and ocular tissues are specified coordinately to ensure directional RGC axon growth is unclear.

Cysts are hollow spheres of a columnar epithelium that is induced from human pluripotent stem cells via embedding hESC sheets in Matrigel; they are used to generate retinal cells (*Kim et al., 2019*; *Lowe et al., 2016*; *Zhu et al., 2013*). Nevertheless, developmental potentials of cysts have not been characterized.

Using the cysts, here we generate CONCEPT telencephalon-eye organoids (*Figure 1A and B*). Cysts were generated as previously described (*Kim et al., 2019*; *Lowe et al., 2016*). The epithelial structure of cysts was demonstrated by apical localization of the TJP1::GFP reporter at the lumen (*Figure 1C*), consistent with previous findings (*Lowe et al., 2016*; *Zhu et al., 2013*). To assess developmental potentials of cysts, individual cysts were manually picked and then seeded onto the Matrigel-coated surface at low densities (*Figure 1A*). After attaching to the culture surface, cysts grew as dome-shaped individual colonies. Subsequent culture of the colonies in a KSR medium (see Materials and methods) led to the self-formation of concentric zones of anterior ectodermal progenitors: an elevated central zone surrounded by multiple zones. This morphology was identifiable under a stereomicroscope (*Figure 1D*) or an inverted microscope. If multiple cysts were fused together or cysts for seeding were too big or small, CONCEPT structures were affected or incomplete. Overall, the concentric structure was abundant.

CONCEPT structures expressed gene markers for telencephalon and eye in concentric patterns. In mice, Foxg1 was specifically expressed in the E10 telencephalon at high levels; it was also expressed in the optic stalk, invaginating optic cup, and lens at lower levels, with the rostral optic stalk connecting the telencephalic vesicle to the invaginating optic cup (*Figure 1E*; *Xuan et al., 1995*); Vsx2 was specifically expressed in the E10 neuroretina (*Figure 1F*); Pax6 was specifically expressed in the E10.5 neuroretina, RPE, lens vesicles, and surface ectoderm (*Figure 1G*; *Liu et al., 2010*). In CONCEPT structures, the expression of FOXG1, VSX2, and PAX6 generally exhibited concentric patterns spanning from the center to the periphery at days 15–17 (*Figure 1H–K*, *Figure 1—figure supplement 1*; a scheme in the left panel in B). At this stage, PAX6 was expressed in multiple concentric zones at distinct levels, with a higher level at a circular zone that also expressed VSX2 (*Figure 1J*, *Figure 1—figure supplement 1*). Concentric zones of FOXG1, VSX2, and PAX6 expression were also found at later stages. At day 26, VSX2 expression expanded peripherally compared to its expression at day 17 (*Figure 1—figure supplement 1C*, *Figure 1—figure supplement 2C*) and largely overlapped with PAX6 expression (*Figure 1—figure supplement 2A–C*). Additionally, VSX2 expression was higher in a zone close to the center and relatively lower in more peripheral zones (*Figure 1—figure supplement*

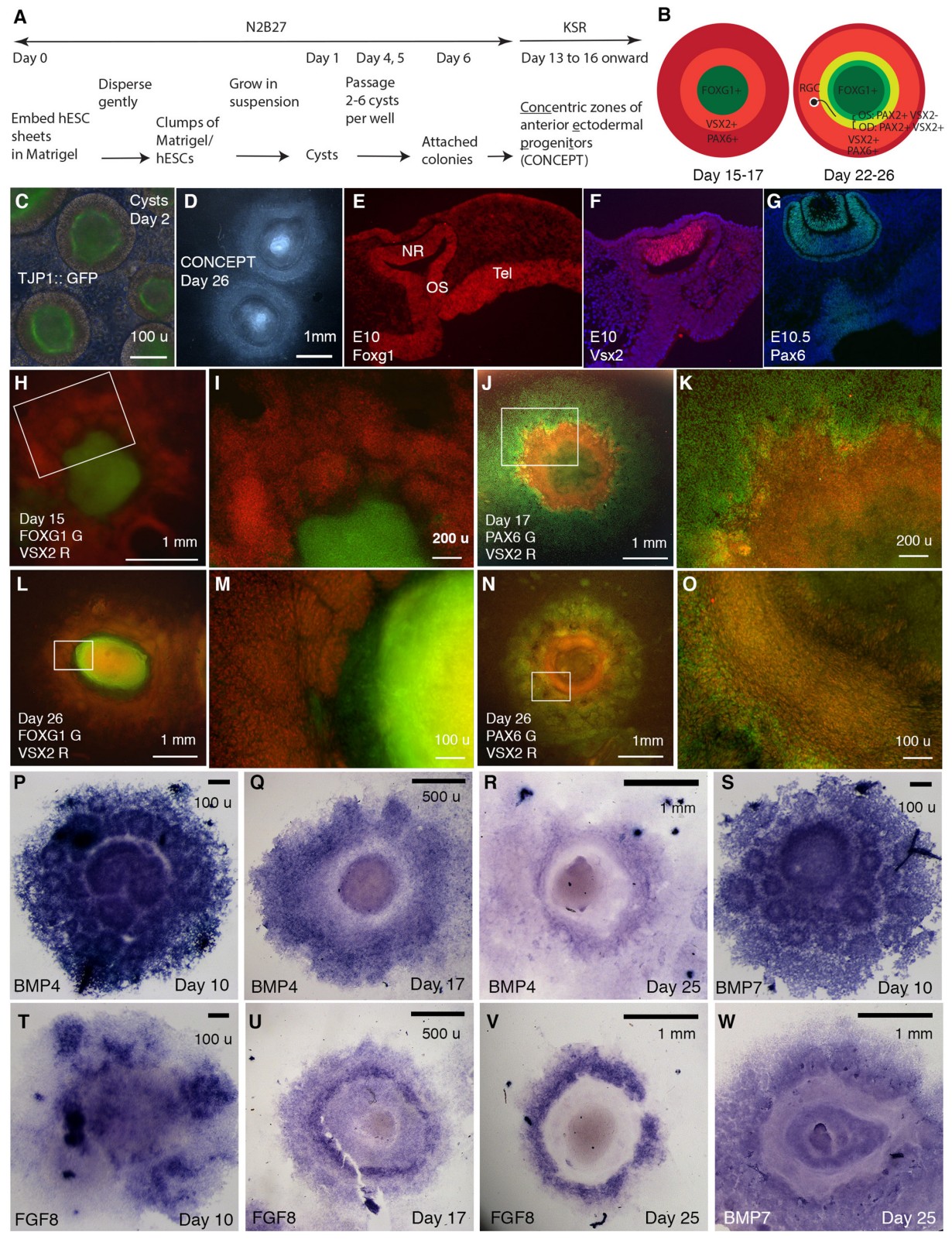

**Figure 1.** Generation of telencephalon-eye organoids comprising *concentric zones* of anterior *ectodermal progenitors* (CONCEPT). (**A**) A scheme of the procedure. (**B**) Diagrams of developing CONCEPT organoids showing concentric zones of the anterior ectodermal progenitors. A summary of *Figures 1, 2 and 7*, *Figure 1—figure supplements 1–3*. (**C**) Morphology of cysts at day 2 showing the epithelial structure indicated by apical localization of the reporter TJ::GFP at the lumen. (**D**) Morphology of CONCEPT organoids at day 26. (**E–G**) Expression of telencephalon (Tel) marker

*Figure 1 continued on next page*

*Figure 1 continued*

Foxg1, neuroretinal (NR) markers Vsx2 and Pax6 in mouse eyes at E10-10.5. Rostral optic stalk (OS) connected the telencephalic vesicle to the optic cup. (**H–O**) FOXG1+ telencephalic progenitors, VSX2+ and/or PAX6+ retinal progenitors formed concentric zones in CONCEPT organoids. N>5 experiments. (**P–W**) In CONCEPT organoids, morphogens FGF8, BMP4, and BMP7 mRNA expression started at early stages and subsequently formed circular gradients. N>5 experiments. Scale bars, 100 μm (**C, E, M, O, P, S, T**), 200 μm (**I, K**), 500 μm (**Q, U**), 1 mm (**D, H, J, L, N, R, V, W**).

The online version of this article includes the following figure supplement(s) for figure 1:

**Figure supplement 1.** Separate-channel images showing VSX2 and PAX6 expression in CONCEPT organoids at day 17.

**Figure supplement 2.** Separate-channel images showing VSX2 and PAX6 expression in CONCEPT organoids at day 26.

**Figure supplement 3.** Reproducibility of CONCEPT telencephalon-eye organoids is demonstrated by consistent gene expression profiles of multiple organoids in whole culture wells.

*2C*). Therefore, the CONCEPT structures comprise spatially organized telencephalic and ocular tissues and thus are named as telencephalon-eye organoids.

Morphogens FGFs and BMPs play crucial roles in patterning the forebrain and eye in vivo. In mice, *Fgf8* is specifically expressed at the rostral forebrain at early stages, induces *Foxg1* expression (*Shimamura and Rubenstein, 1997*), and regulates telencephalic patterning in a dose-dependent manner (*Storm et al., 2006*). *Bmp4* and *Bmp7* are expressed in the dorsomedial telencephalon, optic vesicles, and presumptive lens placodes (*Danesh et al., 2009*; *Dudley and Robertson, 1997*; *Furuta and Hogan, 1998*; *Solloway et al., 1998*). *Bmp4* is required for lens induction (*Furuta and Hogan, 1998*), and *Bmp7* is required for proper patterning of the optic fissure (*Morcillo et al., 2006*). In CONCEPT telencephalon-eye organoids, *FGF8, BMP4,* and *BMP7* were expressed in attached cell colonies starting from early stages (*Figure 1P, S and T*) and subsequently exhibited circular patterns. At day 10, the expression of *BMP4* and *BMP7* delineated multiple rings, with a big ring mostly at the center surrounded by numerous small rings (*Figure 1P and S*). These observations suggest that the attachment of a single-lumen cyst to the culture surface caused differences in cell behaviors: some cells separated from the original cyst and migrated peripherally; some of the separated cells formed small ring-like structures; the cells that remained at the center formed a big ring-like structure. At day 17, circular expression patterns of *BMP4* and *FGF8* emerged (*Figure 1Q and U*). At day 25, *BMP4, FGF8,* and *BMP7* expression clearly marked circular zones (*Figure 1R, V, W*). Expression profiles of *FGFs* and *BMPs* were highly reproducible (*Figure 1—figure supplement 3A–D*). Hence, the expression of *FGFs* and *BMPs* spontaneously form circular gradients, which likely dictate coordinated cell specification in CONCEPT telencephalon-eye organoids.

## Early differentiated RGCs grow directional long axons toward and then along a path defined by PAX2+ VSX2+ cells in CONCEPT telencephalon-eye organoids

To assess cell differentiation in CONCEPT telencephalon-eye organoids, we examined marker expression for RGCs, the first type of cells that differentiate in the neuroretina. In mice, transcription factor Pou4f2 is expressed in the early differentiated RGCs and required for the development of a large set of RGCs (*Gan et al., 1996*). Tubb3 is expressed in the somas and axons of differentiating RGCs.

In CONCEPT organoids, POU4F2 and TUBB3 were detectable as early as day 17 and increased to higher levels at day 22. RGC somas were marked by POU4F2 and TUBB3 co-expression; RGC axons were marked by TUBB3 expression. Interestingly, RGCs grew directional long axons that followed a circular path (*Figure 2A and B*). The circular path of RGC axon outgrowth became more evident in organoids at day 26 (*Figure 2C and D*). In mice, the initially differentiated RGCs are adjacent to the optic disc; their axons grow towards the optic disc to exit the eye and then navigate within the optic stalk to reach their targets in the brain. The optic disc and stalk specifically expressed Pax2 at distinct levels (*Figure 2E–G*), consistent with previous findings (*Bassett et al., 2010*; *Martinez-Morales et al., 2001*; *Mui et al., 2005*). Additionally, presumptive optic disc cells expressed VSX2, whereas optic stalk cells did not (*Figure 1F*, *Figure 2E*; *Liu et al., 2010*). In CONCEPT organoids, there were two PAX2+ cell populations that formed two adjacent rings (*Figure 2H–L*). POU4F2+ RGCs grew TUBB3+ axons toward and then navigated along the adjacent PAX2+ VSX2+ cell population (*Figure 2H–L*), mimicking axon growth from the nascent RGCs toward the optic disc in vivo. Meanwhile, the PAX2+ VSX2- cell population at the inner zone set up an inner boundary of the path for RGC axon growth,

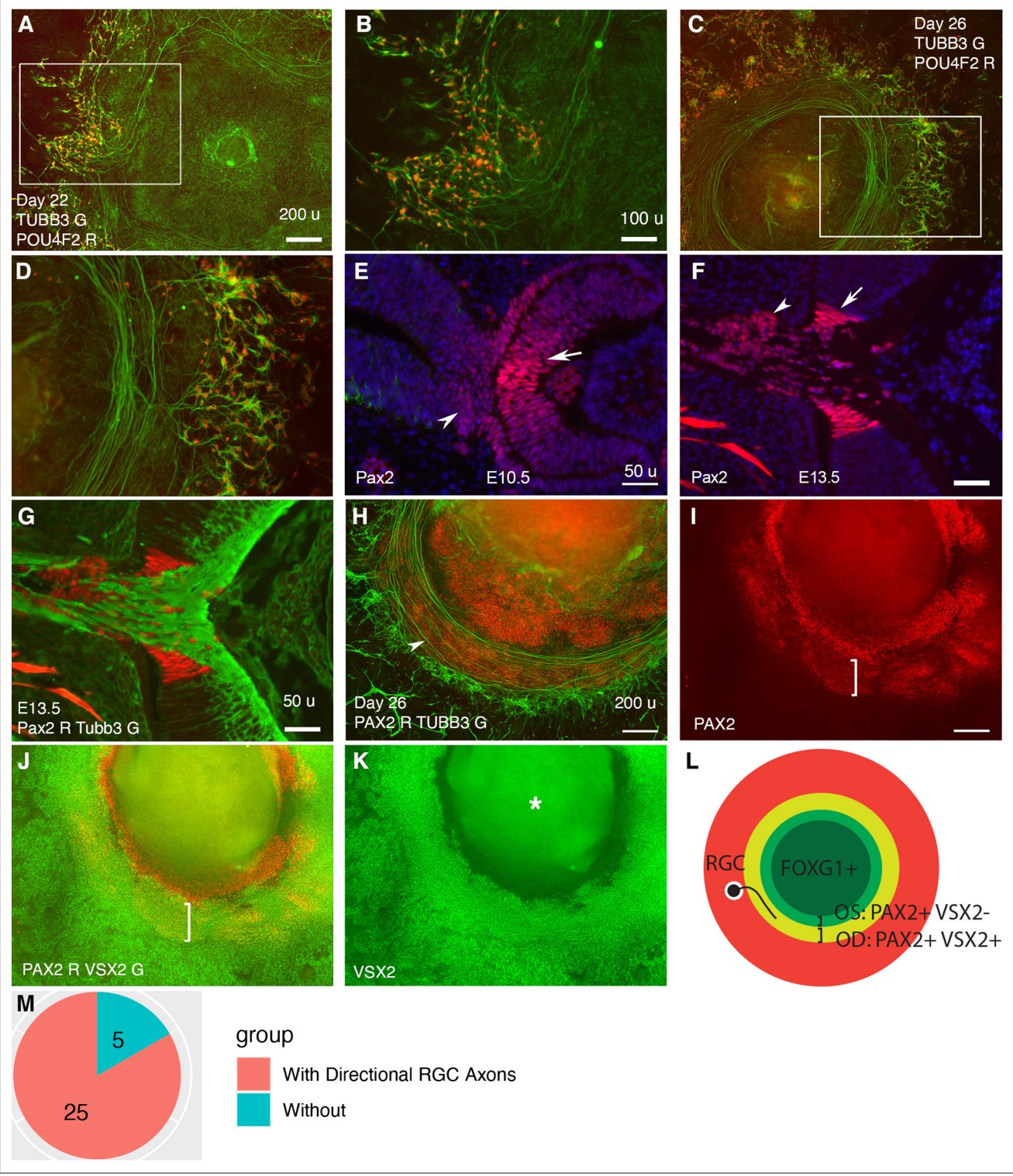

**Figure 2.** Retinal ganglion cells (RGCs) grow axons toward and then along a path defined by PAX2+ VSX2+ cells in CONCEPT telencephalon-eye organoids. N>5 experiments. (**A–D**) POU4F2+ RGCs grew TUBB3+ axons toward and then along a path with a circular or a portion of circular shape. (**E, F**) In mice, Pax2 was expressed in central regions of the retina and optic stalk at E10.5 (**E**) and in the optic disc and optic stalk at E13.5 (**F**). Tubb3+ axons from the initial RGCs grew toward the optic disc, exited the eye, and navigated within the optic stalk (**G**). (**H–L**) In CONCEPT organoids at day 26, TUBB3+ RGC axons grew toward and then along a path defined by an adjacent PAX2+ VSX2+ cell population (arrowhead in H, brackets in I, **J**); the PAX2+ VSX2- cell population set up an inner boundary of RGC axon growth. (**L**) A diagram summarizing RGC axon growth, PAX2+ VSX2+ optic disc (OD), and PAX2+ VSX2- optic stalk (OS) in CONCEPT organoids. The area labeled by the asterisk may appear as false signals in a low-resolution printout

*Figure 2 continued on next page*

Figure 2 continued

but it is clearly a background in digital display. (**M**) A count of CONCEPT organoids showing directional retinal ganglion cell axons. Scale bars, 50 μm (**E, F, G**), 100 μm (**B**), 200 μm (**A, H, I**).

mimicking the optic stalk that spatially confines RGC axon growth in vivo. Based on these findings, we designate PAX2+ VSX2+ cells and PAX2+ VSX2- cells in CONCEPT organoids as optic disc and optic stalk cells, respectively. Taken together, our findings demonstrate that RGCs grow directional axons toward and then along a path defined by PAX2+ VSX2+ cells in CONCEPT telencephalon-eye organoids.

## CONCEPT telencephalon-eye organoids also contain lens cells that undergo terminal differentiation

Besides anterior neuroectodermal cells, we wondered whether CONCEPT organoids also contain non-neural ectodermal cells. CONCEPT telencephalon-eye organoids at around day 25 contained transparent structures reminiscent of the ocular lens. To determine their cell identity, we performed immunostaining. Starting at day 22, lens markers CRYAA and beta crystallin (CRY B) were found (*Figure 3A and B*) and continuously expressed in these transparent cell structures (*Figure 3C, D and K*). Interestingly, these transparent structures were not stained by DAPI (*Figure 3C–F*), indicating denucleation in these lens cells. When CONCEPT organoids were detached using Dispase at around day 28 and continuously grown as suspension cultures, crystal-like clusters with fused transparent spheres—lentoid bodies—were found, and they continuously survived for months (*Figure 3I and J*). Lentoid bodies highly expressed gamma crystallin (CRY G; *Figure 3L*), confirming their lens identity. In mice, terminally differentiated lens fiber cells are free of organelles and featured by specialized interlocking cell membrane domains shown as ball-and-sockets and protrusions (*Biswas et al., 2010*). Using transmission electron microscopy, we found that our lentoid bodies were free of organelles and exhibited ball-and-socket structures (*Figure 3M and N*). Taken together, our findings demonstrate that CONCEPT telencephalon-eye organoids also contain lens cells that undergo terminal differentiation; FGFs in CONCEPT organoids likely promote terminal lens differentiation as seen in other settings (*Lovicu and McAvoy, 2001*).

## Single-cell RNA sequencing analysis identifies telencephalic and ocular cell populations in CONCEPT telencephalon-eye organoids

To fully characterize cell populations in CONCEPT organoids and the mechanisms underlying RGC axon pathfinding, we performed single-cell RNA sequencing (10 x Genomics) of the organoids at day 24, around the stage when RGCs grew long axons toward and along the PAX2+ VSX2+ cell population. Cell Ranger mapping showed that 11158 single cells were sequenced at a depth of 27,842 reads and 2967 genes per cell, and the dataset were further analyzed using Seurat (v3.2.0) (*Stuart et al., 2019*). Cell filtration (nFeature_RNA >200 & nFeature_RNA <6000 & percent.mt <20) resulted in 10,218 cells. Cell clustering grouped cells into 14 clusters (*Figure 4A*; *Table 1*), and cell cycle phases were identified based on cell cycle scores using an established method (*Tirosh et al., 2016*; *Figure 4B*). Several cell clusters, for example, clusters 4, 8, 9, were separated along cell cycle phases (*Figure 4A and B*).

We next assessed cell identities using established markers. Mesoderm, endoderm, and neural crest are identified by a group of gene markers (*Poh et al., 2014*; *Simões-Costa and Bronner, 2015*; *Varga et al., 2021*; *Yao et al., 2017*). In CONCEPT organoids, gene markers for mesoderm (*TBXT, GATA2, HAND1*), endoderm (*GATA1, GATA4, SOX17*), and neural crest (*SNAI1, SOX10, FOXD3*) were not expressed (*Figure 4—figure supplement 1*). In contrast, gene markers for the anterior neuroectoderm were widely expressed. CONCEPT organoids at day 24 were mostly composed of FOXG1+ telencephalic cells (clusters 10, 3, 12, 1, 4, 8, 9, 13) and PAX6+ and/or VSX2+ retinal cells (clusters 2, 7, 5, 6, 11; *Figure 4A–E*). Cluster 0, which comprised both telencephalic and retinal cells (*Figure 4C–E*), was marked by negative gene markers (*Figure 4—figure supplement 2A–D*). Cluster 0, together with cluster 10, had much lower counts of captured RNA and genes compared to other clusters (*Figure 4—figure supplement 2E–G*) and was therefore not dissected further.

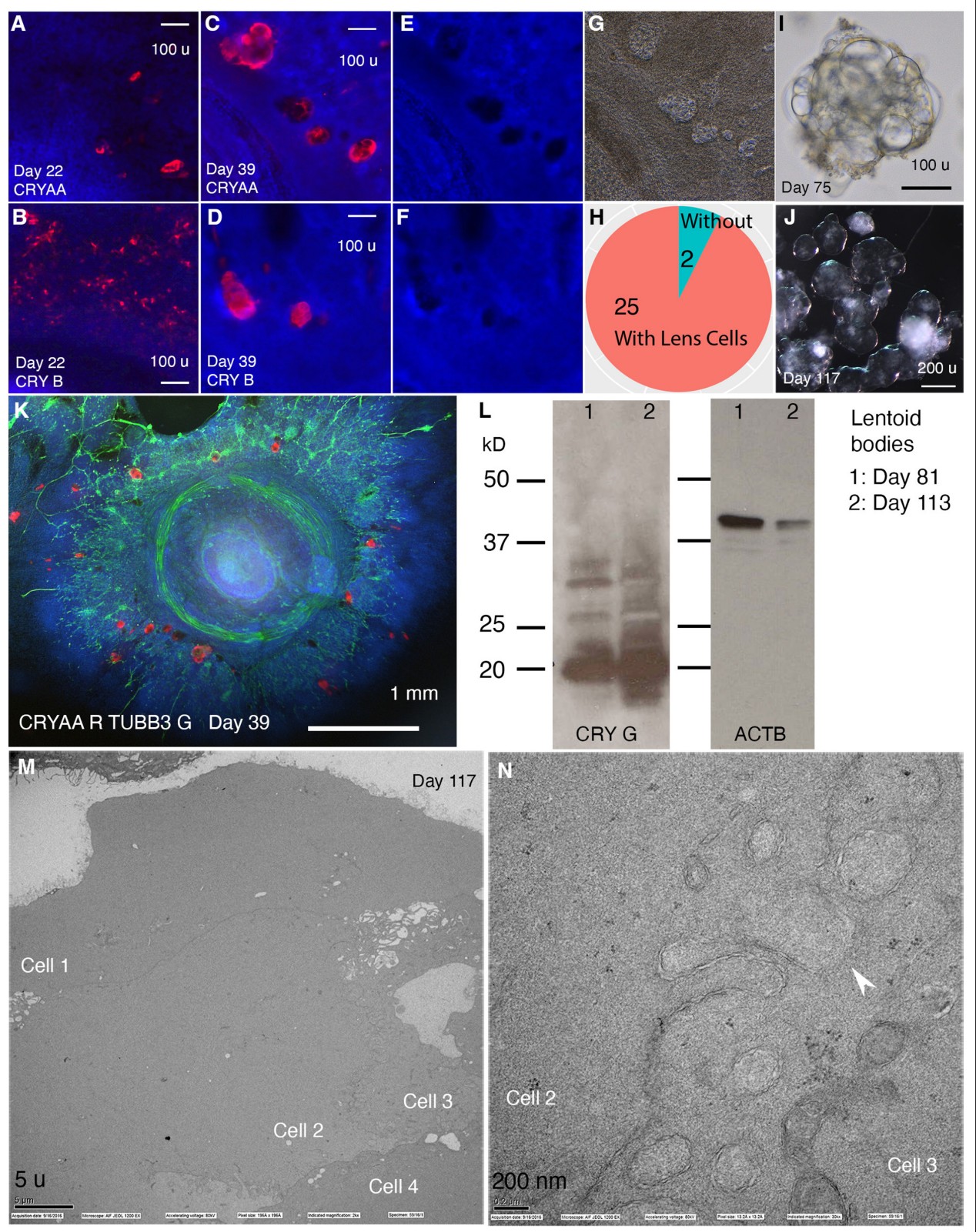

**Figure 3.** CONCEPT telencephalon-eye organoids contain lens cells that undergo terminal differentiation. N>5 experiments. (**A–L**) In CONCEPT organoids, lens markers CRYAA and beta crystallin (CRY B) were expressed at day 22 (**A, B**) and day 39 (**C, D, K**; a count in **H**). Lens cells were not stained by DAPI (**E, F**); they exhibited a crystal-like shape (**G**). When CONCEPT organoids were detached using Dispase at around day 28 and grown in suspension, crystal-like clusters, named as lentoid bodies, were found (**I**) and survived for months (**J**). These lentoid bodies highly expressed gamma

*Figure 3 continued on next page*

*Figure 3 continued*

crystallin (CRY G), as revealed by Western blot (**L**). (**M–N**) These lentoid bodies were free of organelles and exhibited ball-and-socket structures (**K, L**), as revealed by electron microscopy. Scale bars, 100 μm (**A, B, C, D, I**), 200 μm (**J**), 5 μm (**K**), 200 nm (**L**).

In contrast to telencephalon and retinal markers, diencephalon markers (*GBX2, WNT3, SOX14*; *Chatterjee and Li, 2012*; *Martinez-Ferre and Martinez, 2012*) and midbrain/hindbrain markers (*EN2, PAX7, TFAP2B*; *Yao et al., 2017*) were rarely expressed (*Figure 4—figure supplement 3*). Lens markers *CRYAA* and *FOXE3* were expressed in a very small cell population. Those denucleated lens cells were not captured in the scRNA-seq since the absence of nucleus prevented active transcription. Therefore, lens cells did not form a separate cluster probably due to their very low abundance. Taken together, these findings indicate that CONCEPT telencephalon-eye organoids at day 24 are mostly composed of FOXG1+ telencephalic cells and PAX6+ and/or VSX2+ ocular (mostly retinal) cells.

We next characterized FOXG1+ telencephalic cells via assessing differentially expressed genes (DEGs) of clusters. Top DEGs in clusters 4, 8, 9, 1 included *SOX3, FGFR2, PRRX1*, and *EDNRB* (*Figure 4—figure supplement 4A–D*), which mouse orthologs are specifically expressed in the dorsal telencephalon (*Figure 4—figure supplement 5B–E*). Top DEGs in clusters 12, 3, 10 included *DLX2, DLX6-AS1, DLX1*, and *RGS16* (*Figure 4—figure supplement 4E–H*), which mouse orthologs are specifically expressed in the ventral telencephalon (*Figure 4—figure supplement 5F–I*). Consistently, *Foxg1* is expressed in both dorsal and ventral telencephalon in E14.5 mouse embryos (*Figure 4—figure supplement 5A*). Therefore, telencephalic cells in CONCEPT organoids at day 24 comprise both dorsal and ventral telencephalic cells.

Cluster 13 also expressed *FOXG1*, but it was separated from other FOXG1+ cells. Top DEGs in cluster 13 included *TRIB3, DWORF, SLC7A11, GDF15*, and *UNC5B*; cell identities of cluster 13 were undetermined.

PAX6+ and/or VSX2+ retinal cells were grouped into several clusters. *VSX2* was expressed in clusters 2, 7, 5, and 0. Clusters 2 was at G1 phase, cluster 7 was at G1 and S phases, and cluster 5 was at S and G2M phases (*Figure 4A, B and E*). Cluster 6 did not express *VSX2* but expressed *PAX6* and was at the G1 phase (*Figure 4A, B, D and E*). Cluster 6 differentially expressed RPE markers, *e.g., PMEL, HSD17B2, DCT,* and *MITF* (*Figure 4—figure supplement 6*), indicating they were mostly differentiating RPE cells.

Taken together, our single-cell RNA sequencing analysis of CONCEPT telencephalon-eye organoids at day 24 confirms their telencephalic and ocular identities, establishing a valuable transcriptomic dataset for mechanistic studies.

## Identification of two PAX2+ cell populations that mimic to the optic disc and optic stalk, respectively, in the scRNA-seq dataset of CONCEPT telencephalon-eye organoids

To identify PAX2+ cell populations that defined the path for RGC axon outgrowth, we examined *PAX2* expression in the dataset. *PAX2* was mostly expressed in cluster 2 and subsets of clusters 7, 5, 0, 4, 8, and 9 (*Figure 4A and F*). Since clusters 2, 7, 5, and 0 expressed *VSX2* (*Figure 4E and F*), we deduced that PAX2+ VSX2+ cells in cluster 2 and subsets of clusters 7, 5, and 0 corresponded to those PAX2+ VSX2+ cells that defined the path for RGC axon growth (*Figure 2H–L*) and therefore were assigned as optic disc (OD) cells (*Figure 4F*). PAX2+ VSX2+ cells also expressed optic disc and optic stalk marker *SEMA5A* (*Oster et al., 2003*; *Figure 4—figure supplements 7A and 9A, E*). We particularly focused on cluster 2 since PAX2+ VSX2+ cells were mostly found in this cluster. Interestingly, *COL9A3* and *COL13A1* were differentially expressed in cluster 2 (*Figure 4G and H*; *Figure 4—figure supplement 7A*). Previous in situ hybridization results indicate that *COL13A1* is prominently expressed in the optic disc of human fetal retinas (*Sandberg-Lall et al., 2000*). These findings support that PAX2+ VSX2+ cells in CONCEPT telencephalon-eye organoids corresponded to optic disc cells in human fetal retinas. Cluster 2 also differentially expressed glaucoma gene *CYP1B1*, morphogen-encoding genes *LEFTY2, FGF9*, and *FGF8* (*Figure 4I–L*; *Figure 4—figure supplement 7A*). Restricted expression of *FGF8, PAX2, SEMA5A, CYP1B1*, and *LEFTY2* in CONCEPT organoids was validated using in situ hybridization (*Figure 1T–V*; *Figure 4—figure supplement 7B–E*).

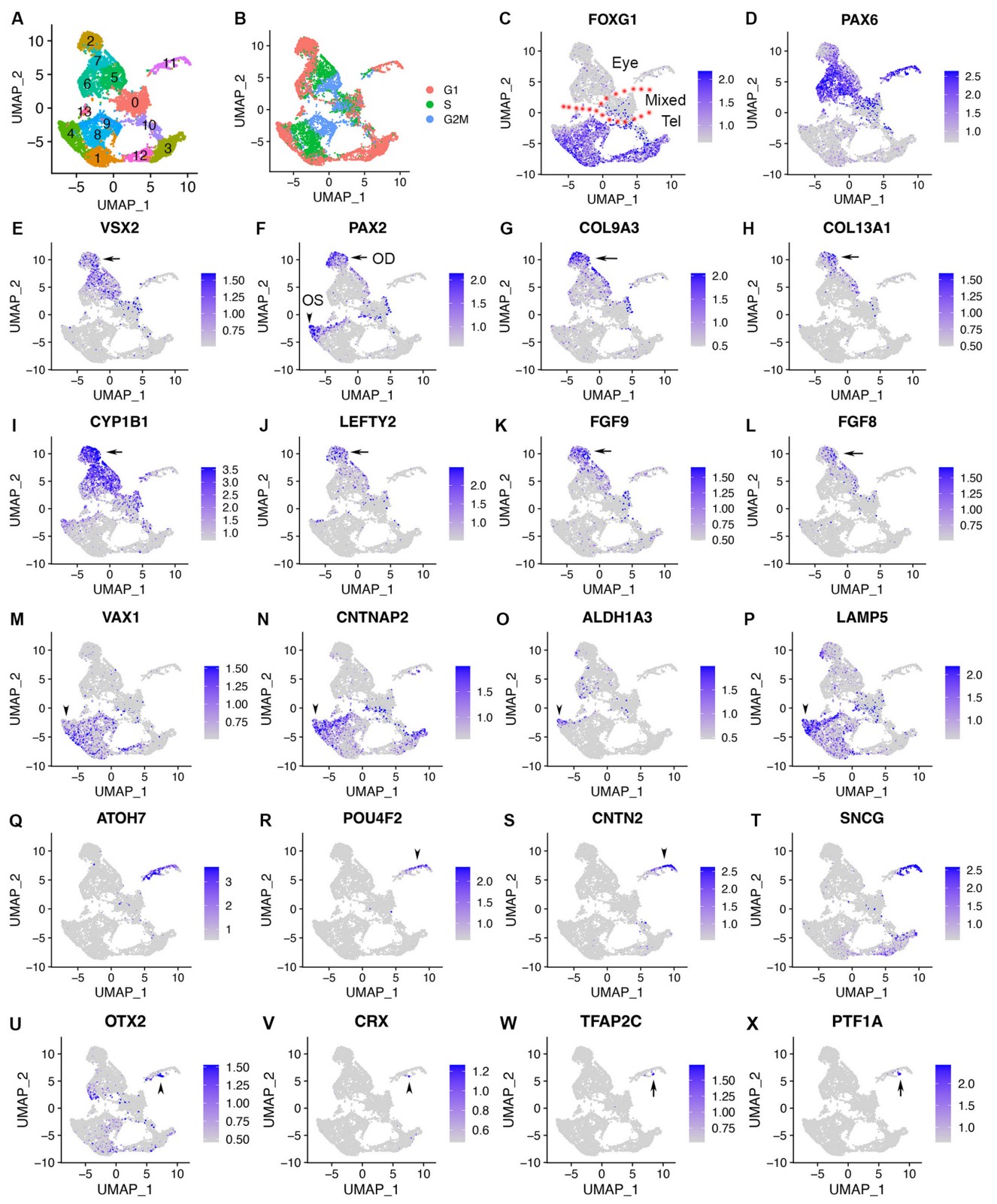

**Figure 4.** scRNA-seq of CONCEPT organoids identifies telencephalic and ocular cells, including PAX2+ VSX2+ optic disc cells, PAX2+ VSX2- optic stalk cells, and CNTN2+ RGCs. CONCEPT organoids at day 24 were used for profiling. (**A**) Identification of 14 cell clusters. (**B**) Cell cycle phases revealed by cell cycle scores. (**C**) *FOXG1* expression marked telencephalic cells. (**D, E**) The expression of *PAX6* and/or *VSX2* marked retinal cells. (**F**) PAX2+ cells were found in two major cell populations: PAX2+ VSX2+ cells were assigned as the optic disc (OD), whereas PAX2+ VSX2- FOXG1+ cells were assigned as the

*Figure 4 continued on next page*

*Figure 4 continued*

optic stalk (OS). (**G–L**) The expression of major DEGs in cluster 2, the major cell population that mimics the optic disc. (**M–P**) The expression of major gene markers for PAX2+ VSX2- optic stalk cells. (**Q–T**) Identification of CNTN2 as a specific marker for early human RGCs. A large portion of cluster 11 differentially expressed neurogenic retinal progenitor marker *ATOH7* and RGC markers *POU4F2* and *SNCG*. The expression of *CNTN2* and *POU4F2* largely overlapped. (**U–X**) Two small portions of cluster 11 differentially expressed early photoreceptor cell markers (*OTX2* and *CRX*, **U, V**) and amacrine/horizontal cell markers (*TFAP2C* and *PTF1A,* **W, X**), respectively.

The online version of this article includes the following figure supplement(s) for figure 4:

**Figure supplement 1.** Mesoderm, endoderm, and neural crest markers are not expressed in CONCEPT telencephalon-eye organoids at 24.

**Figure supplement 2.** Cluster 0 is marked by negative gene markers.

**Figure supplement 3.** Diencephalon markers and midbrain/hindbrain markers are rarely expressed in CONCEPT telencephalon-eye organoids at day 24.

**Figure supplement 4.** DEGs for telencephalic clusters in CONCEPT telencephalon-eye organoids include both dorsal and ventral telencephalic markers.

**Figure supplement 5.** Mouse orthologs of telencephalic DEGs are specifically expressed in dorsal and ventral telencephalon of E14.5 mouse embryos.

**Figure supplement 6.** DEGs in cluster 6 include RPE markers.

**Figure supplement 7.** Expression signatures of cell cluster 2 in CONCEPT organoids.

**Figure supplement 8.** Mouse orthologs of the markers for assigned optic disc and optic stalk cells in CONCEPT organoids are indeed expressed in the optic disc and optic stalk/nerve in E14.5 mouse embryos.

**Figure supplement 9.** Expression of known axon guidance genes *SEMA5A, EFNB2, EFNB1,* and *NTN1* in CONCEPT organoids and at optic disc/stalk areas of E14.5 mice.

**Figure supplement 10.** PAX2+ cells, FOXG1+ cells, SIX3+ cells, and ATOH7+ cells are extremely rare in Gabriel et al.'s organoids.

**Figure supplement 11.** Expression of telencephalic gene marker *FOXG1* and astrocyte markers *OLIG2, CD44,* and *GFAP* in CONCEPT organoids and human fetal retinas HGW9.

In contrast to assigned optic disc cells, PAX2+ VSX2- cells were at edges of telencephalic clusters 4, 8, 9 (*Figure 4A, E and F*) and expressed the optic stalk marker *VAX1* (*Figure 4F and M*). These cells corresponded to those PAX2+ VSX2- cells that set up the inner boundary of the path for RGC axon growth in CONCEPT organoids (*Figure 2H–L*). Therefore, PAX2+ VSX2- cells in subsets of clusters 4, 8, 9 were assigned as optic stalk cells (OS; *Figure 4F*). PAX2+ VSX2- optic stalk cells also differentially expressed *CNTNAP2, ALDHA3,* and *LAMP5* (*Figure 4N–P*), which mouse orthologs are expressed in the optic stalk/nerve (*Figure 4—figure supplement 8*). Compared to assigned optic-disc cells (PAX2+ VSX2+), assigned optic-stalk cells (PAX2+ VSX2-) were closer to telencephalic cells in both the UMAP graph (*Figure 4M–P*; *Figure 4—figure supplement 4A–D*) and their positioning in CONCEPT organoids (*Figure 2H–L*), mimicking the sequence in spatial positions of forebrain, optic stalk, and optic disc in E13.5 mouse embryos.

We also assessed the expression of known axon guidance genes in CONCEPT organoids. *SEMA5a* and *EFNB1* were expressed in both assigned optic disc and stalk cells, *EFNB2* was highly expressed in assigned optic disc cells, and *NTN1* was mostly expressed in assigned optic cells (*Figure 4—figure supplement 9*).

Collectively, we identify two PAX2+ cell populations that mimic the optic disc and optic stalk, respectively, in the scRNA-seq dataset of CONCEPT telencephalon-eye organoids.

**Table 1.** Cell counts for clusters in the scRNA-seq dataset of CONCEPT organoids at day 24.

| Clusters | 0 | 1 | 2 | 3 | 4 | 5 | 6 | 7 | 8 | 9 | 10 | 11 | 12 | 13 |
|---|---|---|---|---|---|---|---|---|---|---|---|---|---|---|
| Idents | Mixed (lower counts) | dTel | OD | vTel | dTel/ OS | NR | RPE | NR | dTel/ OS | dTel/ OS | vTel | RGC/PR/ AC/HC | vTel | UD |
| # Cells | 1295 | 1158 | 964 | 931 | 921 | 871 | 747 | 743 | 729 | 603 | 450 | 393 | 333 | 80 |
| percent | 0.127 | 0.113 | 0.094 | 0.091 | 0.09 | 0.085 | 0.073 | 0.073 | 0.071 | 0.059 | 0.044 | 0.038 | 0.033 | 0.008 |

Abbreviations: Idents, assigned cell identities; # Cells, cell number; Tel, telencephalon; NR, neural retina; OD, optic disc; OS, optic stalk; RPE, retinal pigment epithelial cells; RGC, retinal ganglion cells; PR, photoreceptor cells; AC, amacrine cells; HC, horizontal cells; UD, undetermined.

## Identification of differentiating retinal neurons and RGC-specific cell-surface marker CNTN2

We next assessed cell identities of cluster 11. Interestingly, cluster 11 differentially expressed neurogenic retinal progenitor marker *ATOH7* (*Figure 4Q*), indicating that these cells underwent retinal differentiation. A large portion of cluster 11 differentially expressed *POU4F2* (*Figure 4R*), indicating their RGC identity. Notably, *CNTN2,* which encodes a glycosylphosphatidylinositol (GPI)-anchored cell membrane protein, was largely co-expressed with *POU4F2* (*Figure 4R and S*). Cluster 11 also differentially expressed RGC marker *SNCG* (*Figure 4T*). Two small portions of cluster 11 differentially expressed photoreceptor markers *OTX2* and *CRX* (*Figure 4U and V*) and amacrine/horizontal cell markers *TFAP2C* and *PTF1A* (*Figure 4W–X*), respectively. Therefore, cluster 11 comprises differentiating retinal neurons, predominantly early RGCs; cell surface protein gene *CNTN2* is differentially expressed in early RGCs.

## Transcriptomic comparisons between CONCEPT organoids, brain/optic organoids, and human fetal retinas

We next performed transcriptomic comparisons between CONCEPT organoids, brain/optic organoids, and human fetal retinas. 'Optic vesicle-containing brain organoids' with 'axon-like projections' were recently reported (*Gabriel et al., 2021*). We were curious about the similarities and differences between CONCEPT organoids and Gabriel et al.'s organoids. We loaded Gabriel et al.'s scRNA-seq dataset (*Pasquini, 2021*) and were able to reproduce the cell clustering described in their paper (*Figure 4—figure supplement 10A, C*). Then, we assessed *PAX2* expression in Gabriel et al.'s organoids. Interestingly, PAX2+ cells were extremely rare; there were only eight and five PAX2+ cells scattering across Gabriel et al.'s datasets at day 30 and day 60, respectively (*Figure 4—figure supplement 10B, D*), indicating that Gabriel et al.'s organoids do not have PAX2+ cell clusters that were found in CONCEPT organoids. In Gabriel et al.'s organoids on day 30, FOXG1+ telencephalic progenitors, SIX3+ neuroretinal progenitors, and ATOH7+ neurogenic progenitors were extremely rare (*Figure 4—figure supplement 10E–G*). *VSX2* was not even found in the dataset of Gabriel et al.'s organoids on day 30; *VSX2* was filtered out, probably due to its extremely low expression. Based on these findings, we conclude that PAX2+ optic disc/stalk, FOXG1+ telencephalic, and VSX2+ neuroretinal cell clusters that were found in CONCEPT organoids did not exist in Gabriel et al.'s organoids, indicating striking differences between Gabriel et al.'s organoids and our CONCEPT telencephalon-eye organoids.

Transcriptomic comparisons between CONCEPT organoids at day 24 and human fetal retinas at HWG9 (*Lu et al., 2020*) indicate that CONCEPT organoids and human fetal retinas had similar expression signatures. Cell clustering of HGW9 retinas identified 21 clusters (*Figure 5A*). Notably, cluster 18 differentially expressed *PAX2, COL9A3, CYP1B1, SEMA5A,* and *FGF9* (*Figure 5B–F*), which were top DEGs of cluster 2 in CONCEPT organoids (*Figure 4F, G, I and K*; *Figure 4—figure supplement 9A*). Overall, 64/113 DEGs of cluster 18 in HGW9 were also DEGs of cluster 2 in CONCEPT organoids. In both HGW9 and CONCEPT organoids, expression of *OLIG2, CD44,* and *GFAP* was undetectable (*Figure 4—figure supplement 11*), indicating that astrocytes had not been generated yet at these stages. In the HGW9 dataset, clusters 0–3, 9, 10, 12, 13, 17, 19 highly expressed *VSX2*, indicating that these cells were retinal progenitor cells (*Figure 5G*). Additionally, cluster 18 was a protrusion from VSX2+ cell clusters in the UMAP graph (*Figure 5A–G*), which pattern was similar to the position of cluster 2 relative to VSX2+ clusters in the UMAP graph of CONCEPT organoids (*Figure 4A, E and F*). In the HGW9 dataset, *ATOH7* expression marked neurogenic retinal progenitors (*Figure 5H*), *OTX2* expression marked early photoreceptors (*Figure 5I*), and *TFAP2C* and *PTF1A* expression marked amacrine and/or horizontal cells (*Figure 5J and K*). Notably, *CNTN2* was also largely co-expressed with *POU4F2* in human fetal RGCs (*Figure 5L and M*). Relative positions of ATOH7+, POU4F2+, CNTN2+, OTX2+, TFAP2C+, and PTF1A+ cells in HGW9 and CONCEPT retinas were similar (*Figure 5H–M*, *Figure 4Q, R, S, U, W, X*). When cells in cluster 18 and retinal progenitor clusters from the HGW9 dataset were combined with cells in clusters 2, 4, 5, 7 from the CONCEPT dataset for Seurat anchor-based clustering, cells in cluster 18 from HGW9 (H18) were grouped with cluster 2 from CONCEPT organoids (C2, assigned optic disc; N), and these cells expressed both *PAX2* and *VSX2* (arrowheads in *Figure 5N–R*). A small portion of H18 cells were grouped with cluster 4 from CONCEPT organoids (C4, assigned optic stalk; N), and these cells expressed *PAX2* but not *VSX2* (arrows in *Figure 5N–R*). These findings indicate that most of PAX2+ cells from the HGW9 dataset are optic disc cells, whereas

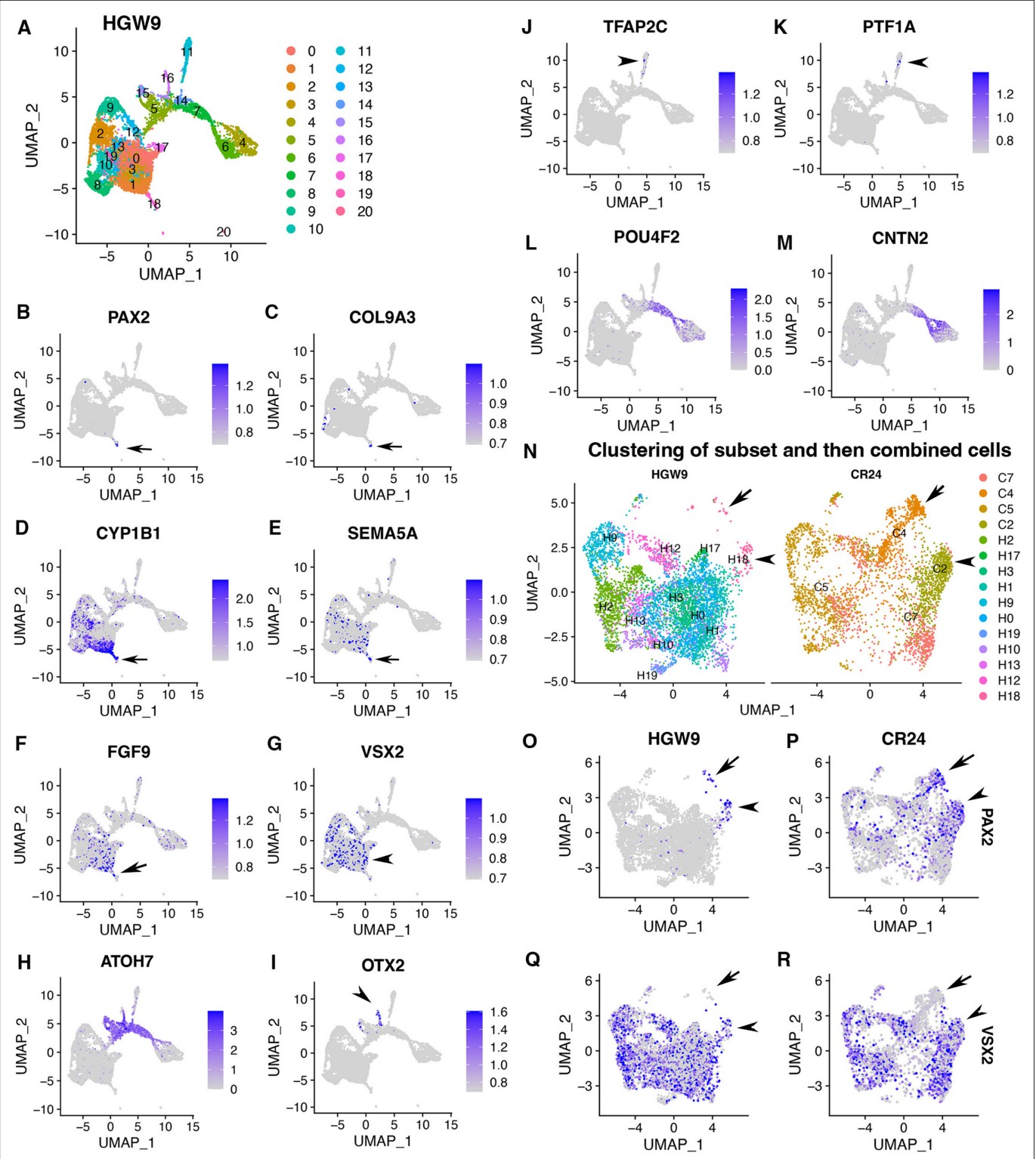

**Figure 5.** Expression signatures of human fetal retinas HGW9 are similar to those of CONCEPT organoids. (**A**) Cell clustering of human fetal retinas HGW9 (GSE138002) identified 21 clusters. (**B–F**) Cluster 18 differentially expressed *PAX2, COL9A3, CYP1B1, SEMA5A,* and *FGF9,* which were top DEGs of cluster 2 in CONCEPT organoids. (**G**) *VSX2* expression marked retinal progenitor cells. (**H–M**) Identification of neurogenic retinal progenitor cells (**H**), early photoreceptor cells (**I**), amacrine/horizontal cells (**J, K**), and early RGCs (**L, M**). *POU4F2* and *CNTN2* were largely co-expressed in early RGCs, consistent with their expression profiles in CONCEPT organoids. (**N–R**) When cells in cluster 18 and retinal progenitors from HGW9 were combined with cells in clusters 2, 4, 5, 7 from CONCEPT organoids (CR24) for Seurat anchor-based clustering, cells in cluster 18 from HGW9 (H18) were grouped with

*Figure 5 continued on next page*

*Figure 5 continued*

cluster 2 from CONCEPT organoids (C2, assigned optic disc; **N**), and these cells expressed both *PAX2* and *VSX2* (arrowheads in N-R). A small portion of H18 cells were grouped with C4 cells (assigned optic stalk; **N**), and these cells expressed *PAX2* but not *VSX2* (arrows in N-R).

a small portion are optic stalk cells, which is consistent with the expectation that optic disc cells are preserved whereas optic stalk cells are mostly lost in whole retina preparations. We would like to point out that human fetal retinas HGW9 were at a more advanced stage than CONCEPT organoids, as indicated by the counts of days and the proportion of RGCs (*Figure 5L and M*; *Figure 4R and S*). It is conceivable that more transcriptomic similarities will be identified when stage-matched samples are used for comparisons.

We then compared functional annotations of DEGs (top 200 genes) of cluster 2 in CONCEPT organoids and DEGs (113 genes) of cluster 18 in human fetal retinas HGW9. Top GO terms in GO:MF, GO:CC, and GO:BP are shown (*Figure 6*). For DEGs of cluster 2 in CONCEPT organoids, top enriched GO terms in GO:MF, GO:CC, and GO:BP were extracellular matrix structural constituent, collagen-containing extracellular matrix, and system development, respectively. Additional interesting GO:BP terms included axon development, astrocyte development, eye development, response to growth factor, cell adhesion, cell motility, neuron projection development, glial cell differentiation, and signal transduction. For DEGs of cluster 18 in human fetal retinas HGW9, top enriched GO terms in GO:MF, GO:CC, and GO:BP were cell adhesion molecule binding, extracellular space, and developmental process, respectively. Many GO terms were enriched in both samples, further indicating transcriptomic similarities in PAX2+ optic disc cells between CONCEPT organoids and human fetal retinas.

Collectively, transcriptomic comparisons indicate that CONCEPT telencephalon-eye organoids are innovative and share similar expression signatures with human fetal retinas, such as specific *CNTN2* expression in early RGCs and co-expression of *PAX2* and *VSX2* in assigned optic-disc cells.

## One-step isolation of developing human RGCs via native marker CNTN2

Differential *CNTN2* expression in early RGCs of both CONCEPT organoids and human fetal retinas caught our attention because CNTN2 is a cell-surface protein that may be used as a native marker for RGC isolation. In literature, Cntn2 is specifically expressed in developing RGCs in mice and chicks (*Chatzopoulou et al., 2008*; *Yamagata and Sanes, 2012*).

In CONCEPT at day 25, CNTN2 exhibited an expression pattern very similar to that of TUBB3 (*Figure 2*, *Figure 7*). CNTN2 was found in the cell membrane of RGCs that expressed POU4F2 in the cell nucleus (*Figure 7A, D*). PAX2+ cells formed inner and outer concentric zones, which were designated as optic-stalk and optic-disc cells, respectively. POU4F2+ RGCs formed a dense circular zone adjacent to PAX2+ optic disc cells; POU4F2+ RGCs were sparse in more peripheral areas (*Figure 7B*). These findings indicate that early differentiated RGCs were adjacent to PAX2+ optic disc cells. Interestingly, RGCs in the dense POU4F2+ zone grew CNTN2+ axons toward and then along the path defined by adjacent PAX2+ optic disc cells (*Figure 7A–E*). In areas where there was a gap in PAX2+ optic disc cells, CNTN2+ axons exited the circular path (diamond arrow in *Figure 7A*; a gap between two arrowheads in *Figure 7B and F*). In regions where POU4F2+ RGCs were a few hundred micrometers away from PAX2+ optic disc cells, RGCs grew axons in centrifugal directions (arrow in *Figure 7C*). Very similar findings were found using hiPSCs (*Figure 7—figure supplement 1*), indicating the reproducibility of CONCEPT organoids in multiple cell lines. PAX2+ optic disc cells did not express ALDH1A3; instead, the path for RGC axon growth was bordered by two cell populations that highly expressed ALDH1A3 (*Figure 7G–H*). In mice, Aldh1a3 expression was low in differentiating RGCs in the central retina but high in peripheral retinal progenitors (*Figure 7K*) and in the optic stalk (*Li et al., 2000*), consistent with ALDH1A3 expression in CONCEPT organoids (*Figure 7G and H*). In mice, Vax1 and Vax2 are expressed in the optic stalk and ventral retina and are jointly required for the optic stalk development (*Mui et al., 2005*; *Take-uchi et al., 2003*). In CONCEPT organoids, cells that set up the inner boundary for RGC axon growth also expressed optic-stalk marker VAX1/2 (*Figure 7I and J*; the antibody recognizes both VAX1 and VAX2). Taken together, immunostaining of CONCEPT organoids confirms that CNTN2 is a specific cell-surface marker for differentiating human RGCs; RGCs grow axons toward and then along a path that is defined by adjacent optic disc cells (PAX2+ VSX2+ ALDH1A3-).

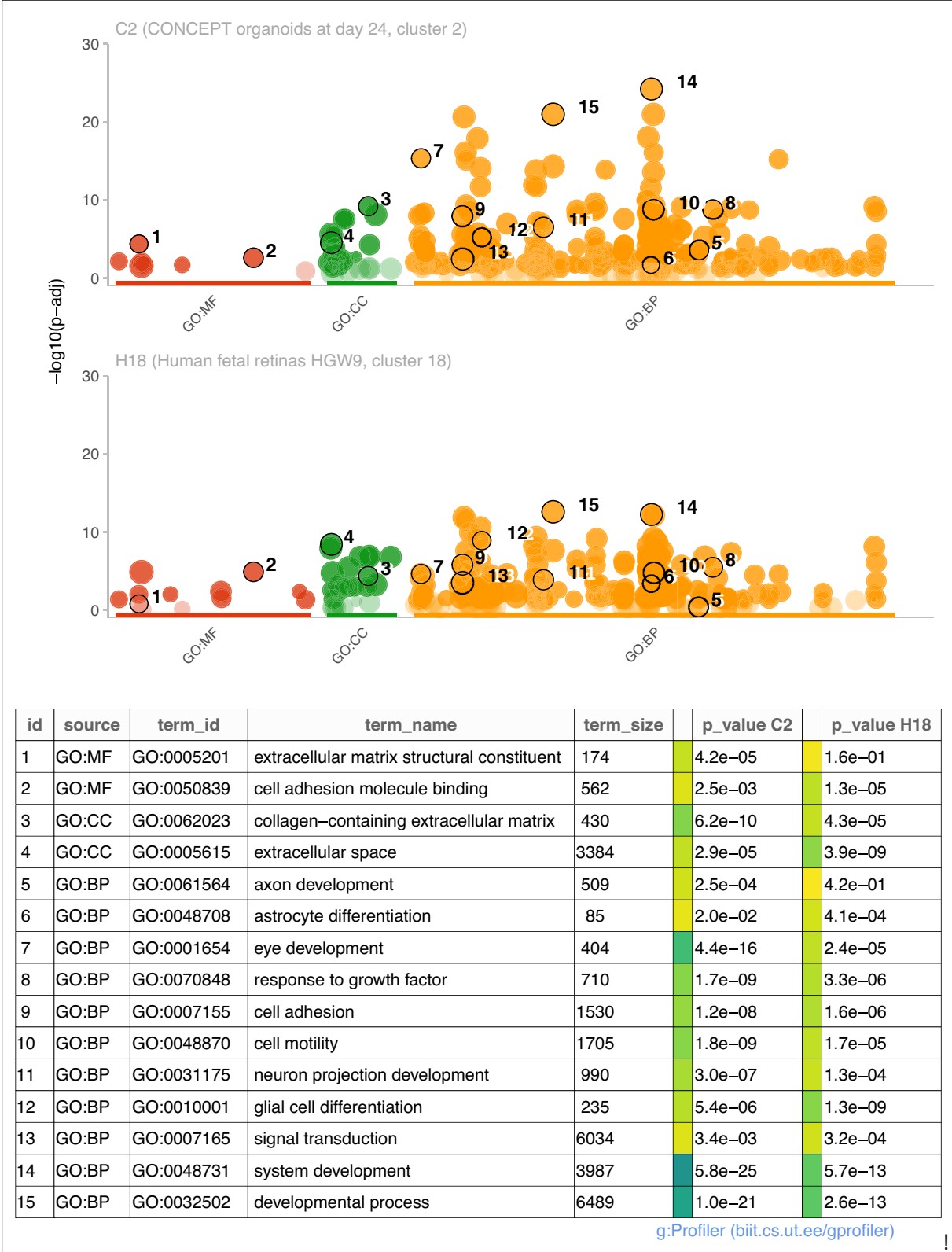

**Figure 6.** Comparisons of enriched GO terms in DEGs (top 200 genes) of cluster 2 in CONCEPT organoids and DEGs (113 genes) of cluster 18 in human fetal retinas HGW9. A large number of GO terms were enriched in both samples.

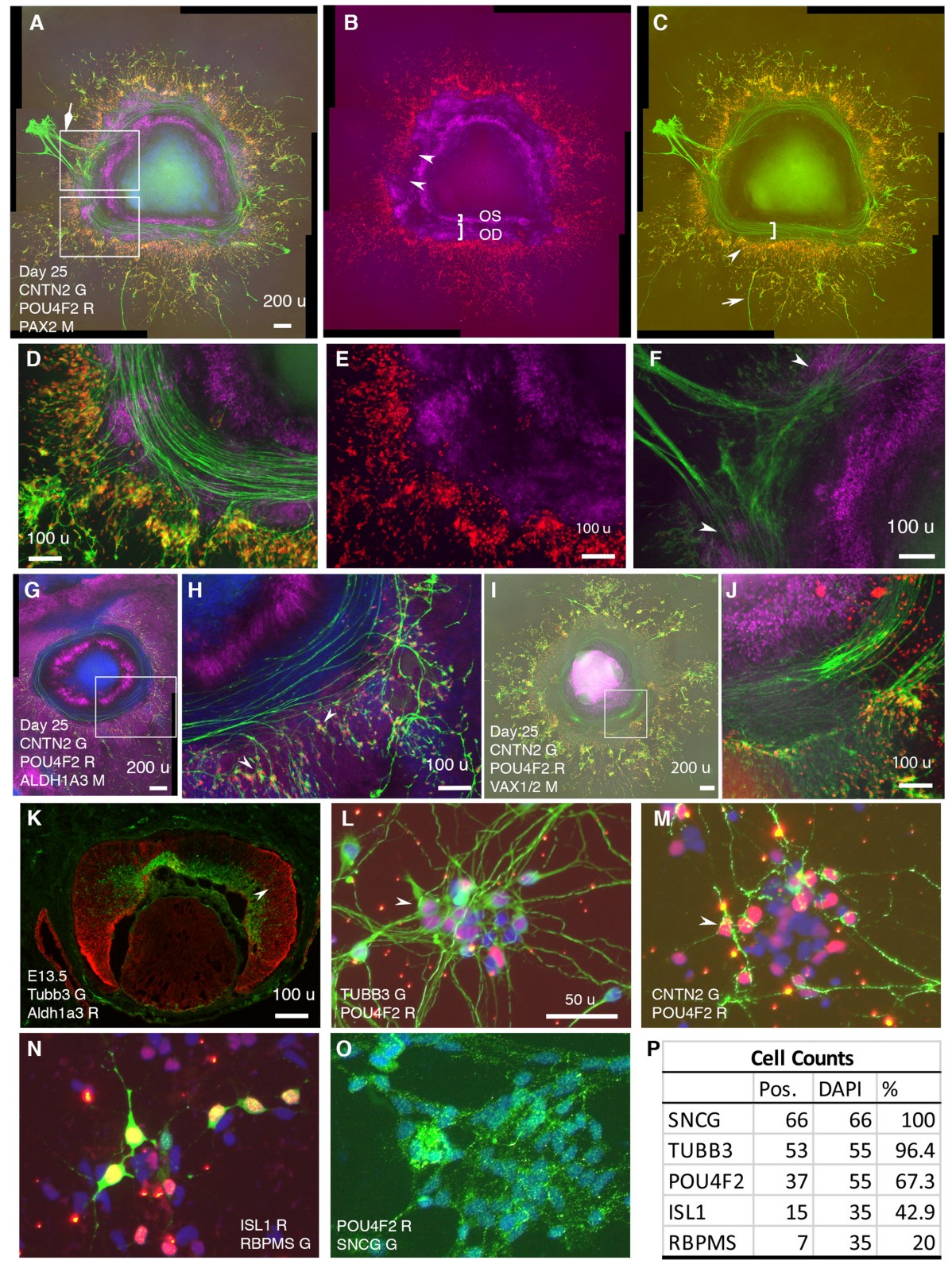

**Figure 7.** RGCs grow CNTN2+ axons toward and then along a defined path in CONCEPT telencephalon-eye organoids and can be isolated in one step via CNTN2 in a native condition. N>5 experiments. (**A–F**) PAX2+ cells formed two concentric zones mimicking the optic stalk (OS) and optic disc (OD), respectively (**A-C**; high magnifications in **D, E**). POU4F2+ RGCs grew CNTN2+ axons toward and then along a path defined by adjacent PAX2+ optic disc cells (**A–E**). RGCs at a few hundreds of micrometers away from PAX2+ optic disc cells grew axons centrifugally (arrow in **C**). At regions where

*Figure 7 continued on next page*

*Figure 7 continued*

there was a gap in PAX2+ optic disc cells, CNTN2+ RGC axons exited the circular path and grew centrifugally (diamond arrowhead in A, double arrowheads in B and F). PAX2+ optic-stalk cells set up an inner boundary for RGC axon growth. (**G, H**) PAX2+ optic disc cells did not express ALDH1A3; the cells that set up the boundaries of the path highly expressed ALDH1A3. (**I, J**) Cells that set up the inner boundary for RGC axon growth expressed VAX1/VAX2 (the antibody recognizes both VAX1 and VAX2). (**K**) In E13.5 mouse eye, Aldh1a3 expression was high in the peripheral retina and was low or nearly absent in the central retina. (**L–P**) One-step isolation of RGCs. RGCs from floating retinal organoids at day 41 (**L, M**) and day 70 (**N–O**) were dissociated into single cells using Accutase and then isolated using MACS via CNTN2 for 10 day growth. Isolated RGCs expressed POU4F2 and grew TUBB3+ neurites in random directions (**L**). RGCs also expressed CNTN2 (**M**), ISL1 (**N**), RBPMS (**N**), and SNCG (**O**); positive cells were counted (**P**). Scale bars, 200 μm (**A,G,I**), 100 μm (**D–F,H,J,K**), 50 μm (**L**).

The online version of this article includes the following figure supplement(s) for figure 7:

**Figure supplement 1.** CONCEPT telencephalon-eye organoids are generated using hiPSC line AICS 0023.

Since cell surface protein CNTN2 was specifically expressed in developing human RGCs, we sought to test whether CNTN2 can be used as a biomarker for isolating human RGCs under a native condition. To that goal, retinal organoids in suspension culture on days 41–70 were dissociated using Accutase to generate a single cell suspension, which was then subject to magnetic-activated cell sorting (MACS) with an antibody against CNTN2. From 100 retinal organoids on days 41–48, around 385,000 RGCs were isolated. Isolated RGCs were plated onto Matrigel-coated chamber slides for 10-day adherent culture. These cells exhibited neuronal morphology and widely expressed RGC markers TUBB3 and POU4F2 (*Figure 7L*). Isolated RGCs tended to form clusters in culture. In contrast to directional axon growth found in CONCEPT organoids, isolated RGCs grew neurites in random directions (*Figure 7L*), indicating the differences in axon pathfinding cues between CONCEPT organoids and purified RGC cultures. RGC neurites were also marked by CNTN2 expression (*Figure 7M*). A portion of isolated RGCs expressed POU4F2 but not CNTN2 (*Figure 7M*), indicating that CNTN2 was downregulated in dissociated RGC cultures. Isolated RGCs also expressed RGC markers ISL1, RBPMS, and SNCG (*Figure 7N–P*). Collectively, we have developed a one-step method for the isolation of differentiating human RGCs in a native condition.

## Isolated RGCs exhibit electrophysiological signature of excitable cells

In order to determine the functional properties of isolated RGCs in culture, we examined their electrophysiological properties using whole-cell patch clamp recordings (*Figure 8*, *see Materials and methods*). Using current-clamp configuration, we found that RGCs displayed a hyperpolarized resting membrane potential (*mean ± SD: –20.1±6.4* mV, *Figure 8A*). When cells were held at –70 mV by current injection, in most cases (6/9), depolarizing currents steps could trigger an action potential, which was followed by a depolarization plateau if the current was injected for more than 10ms (*Figure 8B*). In voltage clamp recordings we examined the nature of the voltage-gated conductances (*see Materials and methods*). From a holding membrane potential of –80 mV, both inward and outward currents ($I_m$) were observed in response to depolarizing voltage steps ($V_m$, *Figure 8C*). Outward currents were primarily mediated by voltage-gated potassium channels, as application of 20 mM TEA significantly reduced their amplitude (*mean $I_K$ ± SD at 60 mV; 964±537 pA in control; 279±201 pA in TEA, U=24.0; p=0.019, two-sided MannWhitneyU test*, *Figure 8D*). Conversely, 1 μM TTX abolished all inward currents, demonstrating that they were mediated by activation of voltage-gated sodium channels (*mean $I_{Na}$ ± SD at –10 mV; –338.2±121 in control; –18.3±13 pA in TTX; U=0.0; p=0.035, two-sided MannWhitneyU test*, *Figure 8E*). Taken together, these results indicate that isolated RGCs exhibit functional features traditionally found in excitable cells, such as in neurons.

## FGF signaling mediated by FGFRs is required for early RGC differentiation and directional axon growth in human cells

Directional RGC axon growth toward and then along PAX2+ VSX2+ optic disc cells in CONCEPT organoids could be mediated by signaling molecules secreted from PAX2+ VSX2+ optic disc cells. scRNA-seq analysis identified expression signatures of cluster 2, the major component of PAX2+ VSX2+ optic disc cells in CONCEPT organoids (*Figure 6*; *Figure 4—figure supplement 7A*). Interestingly, *FGF8* and *FGF9* were differentially expressed in cluster 2 (*Figure 4K, L*; *Figure 9A*; *Figure 4—figure supplement 7*). To assess the roles of FGF8 and FGF9 in CONCEPT organoids, we validated that *FGF8* and *FGF9* were differentially expressed in PAX2+ optic disc cells of CONCEPT organoids

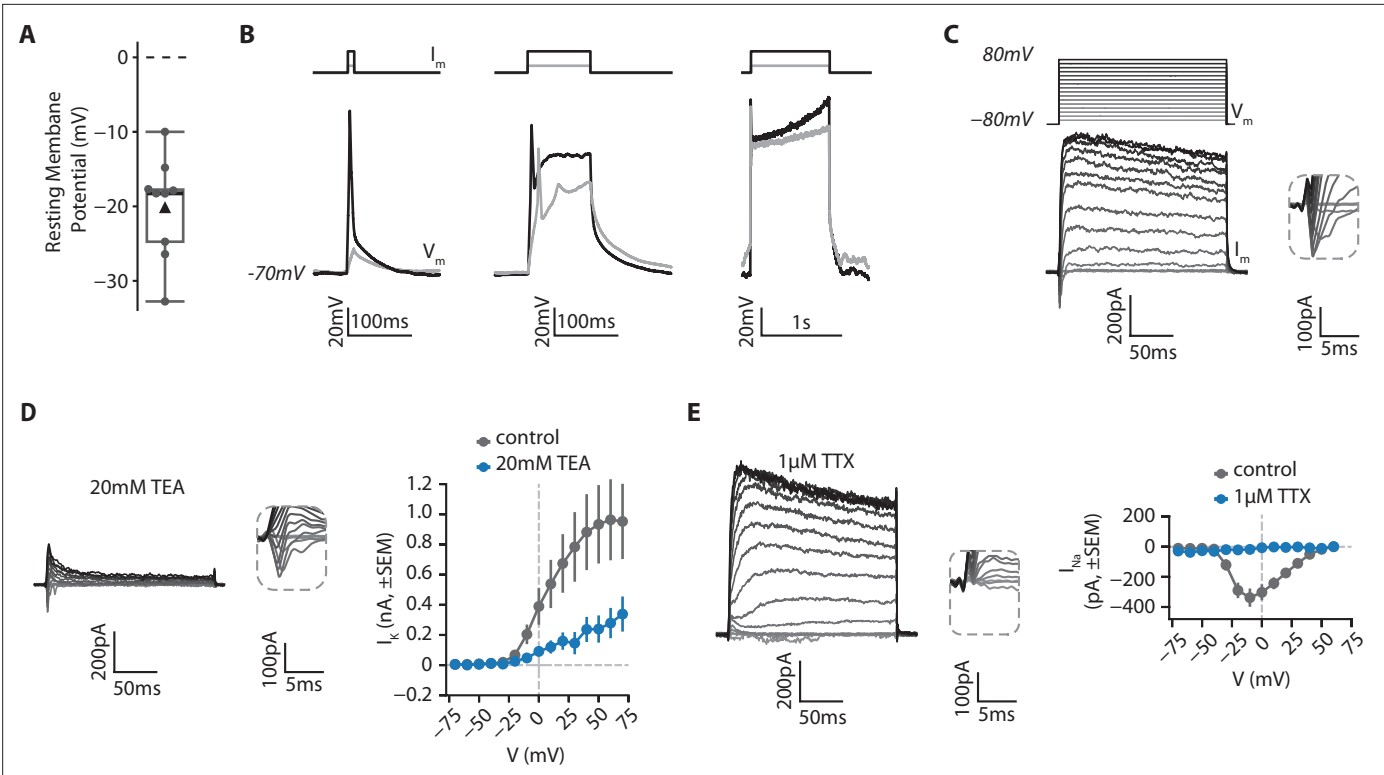

**Figure 8.** Electrophysiological features of RGCs. RGCs from retinal organoids on day 48 were isolated using MACS via a CNTN2 antibody and then grown on polymer coverslips in a chamber slide for 20–25 days before whole-cell patch clamp recordings. (**A**) Resting membrane potential of RGCs. *Black triangle: average; black line: median; box: interquartile range, n=9.* (**B**) RGCs can fire action potentials. Cells were patched in current-clamp mode. A steady current ($I_m$) was injected to maintain the membrane potential at –70 mV and depolarizing current steps of 10ms (*left*), 100ms (*middle*) or 1 s (*right*) were injected to elicit action potentials ($V_m$). (**C**) RGCs show functional voltage-gated currents. Cells were recorded in voltage-clamp (holding=-80mV) and depolarizing voltage steps (200ms, +10 mV steps up to 80 mV, $V_m$) were applied to record inward and outward voltage-gated currents ($I_m$). *Inset*: zoom on inward currents. (**D**) Outward currents are primarily due to voltage-gated potassium channels. *Left*, representative example of a current-voltage experiment performed in presence of 20 mM Tetraethylammonium (TEA), a blocker of voltage-gated potassium channels. *Inset*: zoom on inward currents. *Right*, amplitude of potassium current as a function of membrane potential (*mean ± SEM; $n_{control}$ = 5, $n_{TEA}$ = 5*). (**E**) Inward currents result from activity of voltage-gated sodium channels. *Left*, representative example of a current-voltage experiment performed in presence of 1 µM Tetrodotoxin (TTX), a blocker of voltage-gated sodium channels. *Inset*: zoom on inward currents. *Right*, amplitude of sodium current as a function of membrane potential (*mean ± SEM; $n_{control}$ = 5, $n_{TTX}$ = 3*).

(*Figure 9B–E*). TUBB3+ RGC axons grew towards and then along the regions with high-level *FGF8* and *FGF9* expression (*Figure 9D and E*), suggesting that FGF8 and FGF9 may attract RGC axon growth. Additionally, *FGFR1, FGFR2, FGFR3, MAP2K1*, and *MAP2K2* were expressed in multiple types of cells in CONCEPT organoids. In RGCs, *FGFR1* and *MAP2K2* were clearly expressed (*Figure 9—figure supplement 1*), indicating that RGCs expressed the components that transduce the signals from FGF ligands. Since FGF8 and FGF9 are probably redundant and it is challenging to inactivate both FGF8 and FGF9 in CONCEPT organoids, we decided to inactivate FGF signaling with FGFR1/2/3 inhibitor PD 161570 during days 17–24, a stage at early RGC differentiation. After the FGFR inhibition, FGF8 expression was grossly unaffected. Despite persistent FGF8, the number of RGC somas were drastically reduced (*Figure 9F–K*). Notably, remaining RGCs nearly did not grow directional axons (arrowheads in *Figure 9K*), and a few remaining axons wandered around (arrow in *Figure 9K*). These findings indicate that (a) PAX2+ optic disc cells differentially expressed FGF8 and FGF9; (b) FGF signaling mediated by FGFRs is required for early RGC differentiation and directional axon growth in human cells.

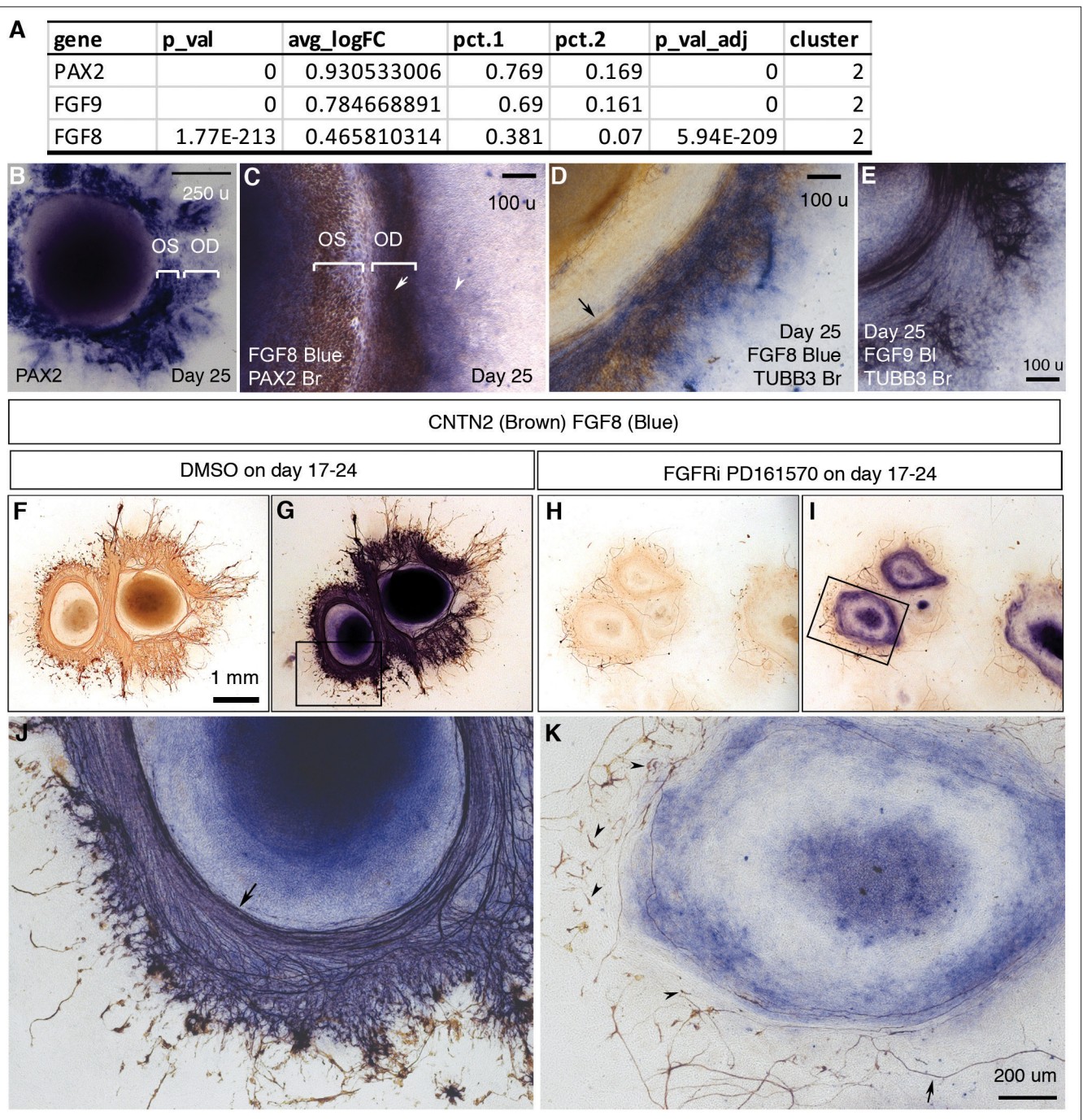

| gene | p_val | avg_logFC | pct.1 | pct.2 | p_val_adj | cluster |
|------|-------|-----------|-------|-------|-----------|---------|
| PAX2 | 0 | 0.930533006 | 0.769 | 0.169 | 0 | 2 |
| FGF9 | 0 | 0.784668891 | 0.69 | 0.161 | 0 | 2 |
| FGF8 | 1.77E–213 | 0.465810314 | 0.381 | 0.07 | 5.94E-209 | 2 |

**Figure 9.** FGF signaling mediated by FGFRs is required for early RGC differentiation and directional axon growth. (**A**) *PAX2, FGF9,* and *FGF8* were differentially expressed in cluster 2, the major component of PAX2+ optic disc cells. (**B**) *PAX2* mRNA expression in CONCEPT organoids on day 25. Two PAX2+ concentric zones corresponding to the optic stalk (OS) and optic disc (OD) are labeled. (**C**) Dual-color immunocytochemistry indicates the co-localization of FGF8 and PAX2 in the optic-disc zone of CONCEPT organoids on day 25. (**D–E**) TUBB3+ axons grew towards and then along the cells that expressed high levels of *FGF8* (**D**) and *FGF9* mRNA (**E**) in CONCEPT organoids on day 25. Immunocytochemistry of TUBB3 was performed after in situ hybridization. (**F–K**) After the inhibition of FGF signaling with FGFR inhibitor PD 161570 during days 17–24, FGF8 expression still largely remained, but the number of RGC somas drastically reduced (**J, K**). Notably, remaining RGCs nearly did not grow directional axons (arrowheads in K), and a few remaining axons wandered around (arrow in K). CNTN2 immunocytochemistry before (**F, H**) and after FGF8 immunocytochemistry (**G, I, J, K**) are shown. N=3/3 experiments. Scale bar, 250 μm (**B**), 100 μm (**C–E**), 1 mm (**F**), 200 μm (**K**).

The online version of this article includes the following figure supplement(s) for figure 9:

**Figure supplement 1.** FGFR1, FGFR2, FGFR3, MAP2K1, and MAP2K2 are expressed in multiple types of cells, including RGCs, in CONCEPT organoids.

## Discussion

In this study, we report the self-formation of concentric zones of telencephalic and ocular tissues in CONCEPT telencephalon-eye organoids from human pluripotent stem cells, establishing a model for studying the early development of telencephalic and ocular tissues in humans. RGCs grew axons toward and then along a path defined by PAX2+ VSX2+ cell populations, setting up a model for studying RGC axon growth and pathfinding. We identified expression signatures of cell clusters in CONCEPT organoids using single-cell RNA sequencing and revealed mechanisms of early RGC differentiation and axon growth. We demonstrated that CONCEPT organoids and human fetal retinas had similar expression signatures. Lastly, we established a one-step method for the isolation of human electrophysiologically-excitable RGCs via CNTN2 under a native condition. Our study not only provides deeper insight into coordinated specification of telencephalic and ocular tissues for directional RGC axon growth in humans, but also generates resources for therapeutic studies of RGC-related diseases such as glaucoma.

### Cysts are hollow spheres of a columnar epithelium mimicking the anterior ectoderm

Cysts are hollow spheres of a columnar epithelium with the apical surface at the lumen. They are induced from pluripotent stem cells via embedding small sheets of hESCs into Matrigel and subsequent growth either in a solid thin film (*Zhu et al., 2013*) or suspensions (*Kim et al., 2019*; *Lowe et al., 2016*). Cyst growth in suspensions is cost-effective and scalable. The formation of cysts induced by Matrigel mimics the epithelization of the epiblast by the extracellular matrix (ECM) in the blastocyst (*Bedzhov and Zernicka-Goetz, 2014*; *Coucouvanis and Martin, 1995*).

In this study, individual cysts efficiently generate telencephalic and ocular tissues in the absence of any extrinsic factors, indicating their default cell fates of the anterior ectoderm. In literature, anterior neural tissues are generated from re-aggregated single pluripotent stem cells through dual inhibition of Smad signaling (*Chambers et al., 2009*) or inhibition of Wnt/ß-catenin signaling (*Nakano et al., 2012*). Besides that, undirected cultures without extrinsic factors are used for the generation of retinal cultures (*Fernando et al., 2022*; *Reichman et al., 2014*; *Zhong et al., 2014*). In our procedure, we did not add any extrinsic factors; the epithelial structure of cysts and subsequent adherent growth at a low density facilitate neural induction and differentiation. Our findings are consistent with the reports that epiblast cells are epigenetically primed for ectodermal fates (*Argelaguet et al., 2019*) and neuroectodermal tissues are the default differentiation from pluripotent stem cells (*Muñoz-Sanjuán and Brivanlou, 2002*; *Tropepe et al., 2001*). Collectively, cysts mimic the anterior ectoderm in the epithelial structure and cell fates.

### Pattern formation and coordinated cell differentiation in CONCEPT telencephalon-eye organoids

Tissue patterning is fundamental for the formation of a body plan, which is defined by the anteroposterior, dorsoventral, and left-right axes. Self-formation of CONCEPT telencephalon-eye organoids establishes an efficient way to pattern early telencephalic and ocular tissues along a spatial axis. In CONCEPT organoids on days 22–26, FOXG1+ telencephalon, PAX2+ VSX2- optic stalk, PAX2+ VSX2+ optic disc, and VSX2+ neuroretinal tissues are positioned along the center-periphery axis, mimicking the sequence of their spatial positions in E13.5 mouse embryos. Our findings indicate that the formation of telencephalic and retinal tissues are highly coordinated, consistent with the prosomere model (*Martínez-Morales et al., 2004*; *Redies and Puelles, 2001*). Collectively, we establish CONCEPT telencephalon-eye organoids for studying the early formation of telencephalic and ocular tissues in humans.

The tissue patterning in CONCEPT telencephalon-eye organoids is originated from the attachment of a radially-symmetric epithelium to the culture surface for growth as colonies since individual cysts kept in suspensions do not form any apparent tissue patterning. When a floating cyst initially contacted the culture surface, the contact resulted in ECM-cell adhesions. The initial ECM-cell adhesions caused additional ECM-cell contacts in neighboring cells, resulting in the flattening and spreading of a cyst onto the culture surface. Since the cyst is a radially symmetric epithelium, ECM-cell adhesions between the cyst and the culture surface formed sequentially in a concentric manner. Remodeling of cell-cell and ECM-cell adhesions and self-organization underlay the tissue

patterning. Molecular events in concentric patterns during the attachment of a cyst onto the culture surface were eventually translated to concentric gradients of morphogens that specify cell fates. The radially symmetric epithelial structure of cysts is important for the formation of a concentric pattern since such pattern would not be generated when amorphous embryoid bodies are attached to the culture surface for growth as colonies. In literature, timed BMP4 treatment is shown to promote neuroretinal differentiation from pluripotent stem cells (*Kuwahara et al., 2015*), and FGF8 promotes telencephalic and eye development (*Martinez-Morales et al., 2005*; *Wilson and Rubenstein, 2000*). In our system, BMP4 and FGF8, along with BMP7 and other FGFs, were highly expressed starting at early stages and gradually formed concentric morphogen gradients, which would dictate tissue patterning, resulting in coordinated specification of telencephalic, optic stalk, optic disc, and neuroretinal tissues along the center-periphery axis in CONCEPT telencephalon-eye organoids. Co-culture of these tissues in the concentric zones provides mutual cues for coordinated tissue specification and growth. Nevertheless, anchorage culture of cell colonies prevents three-dimensional morphogenesis as seen in embryos, impeding proper tissue interactions at more advanced stages. When these cell colonies are detached using dispase for suspension culture, retinal progenitor cells self-organize to form protruding, translucent spheres called retinal organoids, whereas other types of cells form structures with dark appearance; concentric zones no longer exist (*Lowe et al., 2016*). It is conceivable that the simple suspension culture does not have proper physical and chemical supports provided by periocular tissues in vivo and therefore optic-stalk cells do not form a tube-like structure. It is a challenge ahead to reconstruct this three-dimensional environment to mimic the morphogenesis in vivo.

Concentric patterns of stem cell-derived cultures are also reported in a few other experiments. When dissociated single pluripotent stem cells are grown in micropatterned culture surface at certain cell densities in a medium supplemented with BMP4, concentric zones of progenitors expressing markers for trophectoderm, endoderm, mesoderm, and ectoderm are found (*Etoc et al., 2016*; *Minn et al., 2020*). Cell density and colony geometry dictate cell fate specification from pluripotent stem cells. A concentric gradient of BMP4 activity regulates the patterning (*Etoc et al., 2016*). When dissociated single pluripotent stem cells are grown in a pre-patterned geometrically-confined culture surface in a medium supplemented with dual inhibitors for TGF-β and BMP4, concentric zones of progenitors expressing the markers for neural plate and neural plate border are observed. Morphogenetic cues – cell shape and cytoskeletal contractile force –dictate the patterning of the neural plate and neural plate border via BMP-SMAD signaling (*Xue et al., 2018*). When dissociated single pluripotent stem cells are grown as individual colonies in a medium supplemented with knockout serum replacement, multiple zones of ectodermal cells autonomously form (*Hayashi et al., 2016*). In all three experiments, dissociated single cells are used to generate cell colonies through either cell re-aggregation or proliferation. In our case, adherent culture of a radially symmetric epithelium – the cyst – was used to generate CONCEPT organoids. Among the three previous and our experiments, starting cells differ in their developmental potentials, tissue structures, and treatments. Despite the differences, all these organoids comprise concentric zones of distinct cell populations. Formation of CONCEPT telencephalon-eye organoids mimics the coordinated specification of early telencephalic and ocular tissues in humans.

## Signal cues for early RGC differentiation and axon pathfinding in CONCEPT telencephalon-eye organoids and humans

In the mouse retina, multiple RGC axon guidance cues are concentrically organized around the optic disc, regulating RGC axon growth and exit from the eye through the optic stalk (*Oster et al., 2004*). Early differentiated RGCs are in a short distance from the nascent optic disc, and axons of later differentiated RGCs in more peripheral retinal regions follow the path of the initial axons. It is accepted that the optic-disc regions provide growth-promoting guidance cues, whereas peripheral retinal regions provide inhibitory guidance cues (*Oster et al., 2004*). Pax2 is specifically expressed in the ventral optic stalk, optic vesicles, central neuroretina, optic disc, and optic stalk; Pax2 is essential for optic stalk and nerve development in mice (*Macdonald et al., 1997*; *Martinez-Morales et al., 2001*; *Torres et al., 1996*).

In CONCEPT telencephalon-eye organoids, coordinated specification of telencephalic and ocular tissues generates PAX2+ optic disc and optic-stalk cells that define the path for RGC axon growth. To

the best of our knowledge, directional RGC axon growth guided by optic-disc and optic-stalk tissues has not been reported.

Single-cell RNA sequencing of CONCEPT organoids identified DEGs of PAX2+ optic disc and optic-stalk cells. Interestingly, *FGF8* and *FGF9* were differentially expressed in PAX2+ VSX2+ optic disc cells; inhibition of FGF signaling with an FGFR inhibitor during early RGC differentiation drastically decreased the number of RGC somas and nearly ablated directional axon growth, indicating that early RGC differentiation and directional axon growth require FGFR-mediated FGF signaling in human cells. In chicks, FGF8 and FGF3 coordinate the initiation of retinal differentiation *Martinez-Morales et al., 2005*; *Vogel-Höpker et al., 2000*; FGF8 maintains PAX2 expression in optic nerve explants (*Soukkarieh et al., 2007*). In mice, FGF8 and FGF9 trigger axon outgrowth in motor neuron column explants (*Shirasaki et al., 2006*). We conclude that FGF signals provided by optic disc cells regulate early RGC differentiation and directional axon growth in human retinal tissues.

CONCEPT organoids and human fetal retinas shared expression signatures. For example, assigned optic disc cells from both CONCEPT organoids and human fetal retinas differentially expressed *COL9A3, SEMA5A*, and *FGF9*, which may be axon guidance cues for RGC axon growth in humans. Additionally, DEGs of assigned optic disc cells from both CONCEPT organoids and human fetal retinas were enriched for GO terms astrocyte development, neuron projection development, and glial cells differentiation. Furthermore, RGC axon growth and pathfinding guided by optic disc cells were active before astrocytes were generated in both CONCEPT organoids and human fetal retinas. It is conceivable that CONCEPT organoids are a testable model for studying the functions of axon guidance molecules that are identified using scRNA-seq datasets.

### One-step isolation of early human RGCs under a native condition for both organoids and human fetal retinas

RGCs are degenerated in glaucoma, a major cause of vision impairment in developed countries. Disease modeling and drug discovery to suppress RGC death will have a huge impact on saving vision. The use of human RGC models is critical for therapeutic studies since humans and rodents differ significantly in RGCs. Additionally, cell replacement therapies for glaucoma are extensively evaluated. Therefore, efficient isolation of human RGCs in a native condition will have a substantial impact on therapeutic studies of RGC-related diseases such as glaucoma. In literature, cell surface marker Thy1 is used for the isolation of adult mouse RGCs, but it is unsuccessful in the isolation of RGCs from 3D retinal organoids (*Sluch et al., 2017*). A study reports the purification of human RGCs via Thy1 from adherent cultures (*Rabesandratana et al., 2020*). Tagging human RGCs with an engineered marker Thy1.2 leads to efficient isolation of human RGCs (*Sluch et al., 2017*), but may not be unsuitable for clinical uses. Using single-cell RNA sequencing, we identified cell-surface protein CNTN2 as a specific marker for early RGCs in both human organoids and fetal retinas. We isolated electrophysiologically excitable RGCs from retinal organoids using MACS via CNTN2, and it is conceivable that this method is applicable for human fetal retinas. Therefore, we establish a one-step method for isolating early human RGCs under a native condition, facilitating therapeutic studies for RGC-related retinal diseases such as glaucoma.

# Materials and methods

**Key resources table**

| Reagent type (species) or resource | Designation | Source or reference | Identifiers | Additional information |
|---|---|---|---|---|
| Cell line (*Homo-sapiens*) | H1 hESCs | WiCell | WA01 | |
| Cell line (*Homo-sapiens*) | hiPSCs | Corriell Institute | AICS 0023 | |
| Antibody | anti-FOXG1 (Rabbit polyclonal) | Abcam | Cat# ab18259 | IF (1:500) |
| Antibody | Anti-TUBB3 (mouse monoclonal) | Covance | Cat# MMS-435P | IF (1:1000) |
| Antibody | Anti-FGF8 (mouse monoclonal) | R&D | Cat# MAB323 | ICC (1:500) |

*Continued on next page*

*Continued*

| Reagent type (species) or resource | Designation | Source or reference | Identifiers | Additional information |
|---|---|---|---|---|
| Antibody | Anti-RBPMS (rabbit polyclonal) | PhosphoSolution | Cat# 1830-RBPMS | IF (1:200) |
| Antibody | Anti-ISL1 (mouse monoclonal) | DSHB | 40.2D6 | IF (1:500) |
| Antibody | Anti-SNCG (rabbit polyclonal) | Abcam | Cat# ab55424 | IF (1:200) |
| Antibody | Anti-PAX2 (rabbit polyclonal) | Invitrogen | Cat# 716000 | IF (1:200) |
| Antibody | Anti-alpha A crystallin (rabbit polyclonal) | Santa Cruz sc | Cat# sc-22743 | IF (1:500) |
| Antibody | Anti-beta crystallin (rabbit polyclonal) | Santa Cruz | Cat# sc-22745 | IF (1:100) |
| Antibody | Anti-gamma crystallin (rabbit polyclonal) | Santa Cruz | Cat# sc-22746 | Western (1:1000) |
| Antibody | Anti-CNTN2 (mouse monoclonal) | DSHB | Cat# 4D7 | IF (1:100) |
| Antibody | Anti- ALDH1A3 (rabbit polyclonal) | Invitrogen | Cat# PA529188 | IF (1:500) |
| Antibody | Anti-VAX1/2 (rabbit polyclonal) | Santa Cruz | Cat# sc-98613 | IF (1:200) |
| Antibody | Anti-PAX6 (rabbit polyclonal) | Covance | Cat# PRB-278P | IF (1:500) |
| Antibody | Anti- POU4F2 (goat polyclonal) | Santa Cruz | Cat# SC-6026 | IF (1:200) |
| Antibody | Anti-VSX2 (sheep polyclonal) | Millipore | Cat# AB9016 | IF (1:500) |
| Commercial assay or kit | DIG RNA Labeling Mix | Millipore Sigma | Cat# 11277073910 | |
| Commercial assay or kit | MagnaBind goat anti-mouse IgG beads | ThermoScientific | Cat# 21354 | |
| Sequence-based reagent | BMP4, forward | This paper | PCR primers | CGGAAGCTAGGTGAGTGTGG |
| Sequence-based reagent | BMP4, reverse | This paper | PCR primers | GAGtaatacgactcactatagggGGAAGCCCCTTTCCCAATCA |
| Sequence-based reagent | BMP7, forward | This paper | PCR primers | gaggtccctctccattccct |
| Sequence-based reagent | BMP7, reverse | This paper | PCR primers | GAGtaatacgactcactatagggtgcacccatcagacctccta |
| Sequence-based reagent | FGF8, forward | This paper | PCR primers | GTTGCACTTGCTGGTCCTCT |
| Sequence-based reagent | FGF8, reverse | This paper | PCR primers | GAGtaatacgactcactatagggTTGAGTTTTGGGTGCCCTAC |
| Sequence-based reagent | PAX2, forward | This paper | PCR primers | gctgtctgtgctgtgagagt |
| Sequence-based reagent | PAX2, reverse | This paper | PCR primers | GAGtaatacgactcactatagggccggggacatttagcaggtt |
| Sequence-based reagent | SEMA5A, forward | This paper | PCR primers | CAGAGGCTCAGGCACAATGA |
| Sequence-based reagent | SEMA5A, reverse | This paper | PCR primers | GAGtaatacgactcactatagggTCCGTGTCTACCCAGGACTT |
| Sequence-based reagent | CYP1B1, forward | This paper | PCR primers | cccagcggttcttcatgagt |
| Sequence-based reagent | CYP1B1, reverse | This paper | PCR primers | GAGtaatacgactcactatagggggcacacttggttgcgttagt |
| Sequence-based reagent | LEFTY2, forward | This paper | PCR primers | agccctctaactgaacgtgtg |
| Sequence-based reagent | LEFTY2, reverse | This paper | PCR primers | GAGtaatacgactcactatagggtcttctgagtatctacattcaattgct |
| Sequence-based reagent | EMX2, forward | This paper | PCR primers | ACCGAGAAAGGGAGAGGGAA |
| Sequence-based reagent | EMX2, reverse | This paper | PCR primers | GAGtaatacgactcactatagggTCGGCCAATTTCTCCAACCA |
| Sequence-based reagent | FGF9, forward | This paper | PCR primers | GTCCGCTATGAACCTGTGGT |
| Sequence-based reagent | FGF9, reverse | This paper | PCR primers | GAGtaatacgactcactatagggATAGTCTCGCTTGCCCAAGG |

*Continued on next page*

*Continued*

| Reagent type (species) or resource | Designation | Source or reference | Identifiers | Additional information |
|---|---|---|---|---|
| Chemical compound, drug | PD 161570 | Tocris | Cat# 3724 | 1 μM |
| Software, algorithm | Seurat | *Stuart et al., 2019*. | Seurat v3.2.0 | |
| Software, algorithm | CellRanger | 10 x Genomics | CellRanger (3.1.0) | |

## Maintenance of cell lines

H1 human pluripotent stem cells from WiCell were used in this study. The cells were authenticated using the expression analysis of pluripotent gene markers and were free from mycoplasma contamination. ESCRO and IRB committees at AECOM approved the use of hESCs in this project. Undifferentiated H1 hESCs (WiCell WA01) or hiPSCs (Corriell Institute AICS 0023) were grown on Matrigel-coated six-well plates in mTeSR1 medium and passaged using ReLeSR (STEMCELL technologies) following manufacturer instructions.

## Retinal cell differentiation

CONCEPT telencephalon-eye organoids were generated as follows. A humidified incubator at 37 °C with 5% $CO_2$ was used for cell culture. H1 hESCs or iPSCs that were passaged using ReLeSR 2 or 3 days before experiments were detached using Dispase (GIBCO 17105041) and then harvested using centrifugation. After that, the cell pellets were suspended in ice-cold Matrigel. After gelling at 37 °C for 15–20 min, the hESC/Matrigel clump was gently dispersed in a N2B27 Medium (DMEM/F12+GlutaMAX (GIBCO):Neurobasal medium (GIBCO)=1:1, 0.5 x B27 supplement (GIBCO), 0.5 x N2 supplement (GIBCO), 0.1 mM β-mercaptoethanol, and 0.2 mM L-GlutaMax) for floating culture. With the starting day of cell differentiation designated as day 0, cysts with a single lumen formed on day 1. Cysts with a single lumen constituted over 85% of cultures. On days 4 or 5, individual cysts with a diameter at around 150–200 μm were manually picked using a curved Pasteur pipet under an inverted microscope and then seeded onto Matrigel-coated 24-well plates at a density of 2–6 cysts per well. Cysts spontaneously attached to the culture surface and grew. From a time during days 13–16, attached cell colonies were grown in a KSR medium (GMEM medium supplemented with 10% knockout serum replacement, 1 mM sodium pyruvate, 0.1 mM non-essential amino acids, 2 mM l-glutamine, and 55 μM 2-mercaptoethanol; all the culture reagents were from Life Technologies). Culture mediums were changed every two or three days. Overall, the procedure is efficient and robust. Images in the manuscript represent 80–90% of cultures.

## Inhibition of FGF signaling in CONCEPT telencephalon-eye organoids with FGFR inhibitor PD 161570

To inactivate FGF signaling in CONCEPT organoids, FGFR1/2/3 inhibitor PD 161570 (1 μM; Tocris) was supplemented to the culture medium starting on day 17, with vehicle DMSO as a control. Treated CONCEPT organoids were harvested on day 24 for assays.

## Magnetic-activated cell sorting (MACS) of developing human RGCs

Retinal organoids in suspension culture were generated using the established method (*Kim et al., 2019*; *Lowe et al., 2016*). For each MACS experiment, 84–140 retinal organoids at stages of days 41–70 were dissociated into single cells using Accutase (GIBCO A1110501). Non-retinal cells were trimmed if there were any. Dissociated single cells were harvested using centrifugation and then incubated for 25 min at room temperature with MagnaBind goat anti-mouse IgG (ThermoScientific 21354) beads that were previously coupled with a CNTN2 antibody (DSHB 4D7) following manufacturer instructions. Cells bound to the beads were isolated using a magnetic rack and then washed one time with the KSR medium supplemented with antibiotic:Antimycotic (GEMINI 400101) while the tube was still against the magnetic rack. After the wash, the cells were released from the beads via Accutase digestion for 30 min and then harvested using centrifugation. The isolated cells were plated onto a chamber slide (ibidi 80826, ibidiTreat μ-Slide 8 Well, coated with poly-ornithine and Matrigel, 30,000–50,000 cells/200 μl/well) in BrainPhys neuronal medium (Stem Cell Technology 05790) supplemented with N2 and B27 (GIBCO 17502001, A3582801). From 100 retinal organoids on days 41–48,

around 385,000 RGCs were isolated. After 10 days of culture in chamber slides, RGCs were fixed in 4% paraformaldehyde (PFA) for 10 min and then processed for immunostaining.

## Immunostaining, antibodies, and light microscopy

CONCEPT telencephalon-eye organoids were fixed in 4% PFA for 15–30 min at room temperature and processed for immunostaining. The detailed information on primary antibodies is found in the Key Resources Table. Primary antibodies were visualized using Alexa Fluor 488-, 568-, and 647-conjugated secondary antibodies and imaged using a Zeiss AxioObserver Z1 microscope. When the sample did not fit in one image, multiple images were stitched to obtain an overview. In dual-color immunocytochemistry for PAX2 and FGF8, FGF8 was visualized using AP-conjugated anti mouse secondary antibody (Invitrogen A16038), and PAX2 was detected by biotin-conjugated anti rabbit secondary antibody (Invitrogen B2770) followed by HRP-conjugated streptavidin (Thermo Fisher Scientific 21130). For dual color immunocytochemistry of CNTN2 and FGF8 (both are mouse antibodies; no working antibodies raised in different species are available), immunocytochemistry of CNTN2 was performed first and visualized using HRP-conjugated streptavidin. Then, the samples were fixed in 4% PFA for 15 min to avoid antibody crosstalk. After that, FGF8 antibody was applied and then detected by an AP-conjugated secondary antibody. Brown and blue colors are well separated in *Figure 5K*, indicating that antibody crosstalk did not exist.

## In situ hybridization

DIG-labeled anti-sense RNA probes for in situ hybridization were generated via in vitro transcription using a DIG RNA labeling kit (Millipore Sigma-Aldrich 11175025910). Sequences of PCR primers for the generation of probe templates are found in the Key Resources Table. To assess the in vivo expression of DEGs identified by single-cell RNA sequencing, in situ hybridization images of their mouse orthologs in E14.5 mouse brains were downloaded from a public database (*Diez-Roux et al., 2011*; *Visel et al., 2004*) (https://gp3.mpg.de/) with permission and then assembled in supplemental figures.

## Electron microscopy (EM)

EM was performed by Analytical Imaging Facility at Albert Einstein College of Medicine with a standard method. Lens cell clusters were fixed in 0.1 M Cacodylate buffer containing 2% paraformaldehyde and 2.5% glutaraldehyde for 60 min at room temperature and then processed for EM.

## Single-cell RNA sequencing

CONCEPT telencephalon-eye organoids at day 24 (referred to as CR24) from one culture well were dissociated into single cells using activated Papain (Worthington Biochemical Corporation LS003126) following manufacturer's instructions. Then, 10,000 dissociated cells were captured using Chromium Controller (10 x Genomics), followed by library preparation using Single Cell 3' version 3.1 kit (10 x Genomics) per manufacturer's instructions. The library was sequenced in one lane of Illumina HiSeq (2x150 bp) by GeneWIZ company.

Fastq sequences were mapped to the human genome (GRCh38-3.0.0) using CellRanger (3.1.0) to generate a count matrix, which was further analyzed using the Seurat Package (v3.2.0; *Stuart et al., 2019*). Sequenced cells were filtered (nFeature_RNA >200 & nFeature_RNA <6000 & percent. mt <20), resulting in 10,218 cells. Dimension reduction and clustering were performed using the following functions: *NormalizeData, FindVariableFeatures (selection.method = "vst", nfeatures = 2000), ScaleData, RunPCA, ElbowPlot, FindNeighbors (dims = 1:17), FindClusters (resolution = 0.5), and RunUMAP(dims = 1:17).* Differentially expressed genes were identified using the function *FindAllMarkers (only.pos=FALSE, min.pct=0.25, logfc.threshold=0.25).* Cells in human fetal retinas dataset HGW9 (GSE138002) were also clustered using the Seurat package, with filtration of nFeature_RNA >200 & nFeature_RNA <5000 & percent.mt <5. For integration of the datasets of CONCEPT organoids and human fetal retinas HGW9 (GSE138002), cells in cluster 18 and retinal progenitor clusters from the HGW9 dataset were combined with cells in clusters 2, 4, 5, 7 from the CONCEPT dataset for Seurat anchor-based clustering using functions *FindIntegrationAnchors* and *IntegrateData.* The integrated dataset was then used for cell clustering. For enriched GO term analysis, DEGs (top 200 genes) of cluster 2 in CONCEPT organoids and DEGs (113 genes) of cluster 18 in human fetal retinas HGW9 were used as inputs for running the function *gost* in the gprofiler2 package (*Kolberg et al., 2020*).

## Whole-cell patch clamp recordings of isolated RGCs in culture

RGCs were isolated from retinal organoids on day 48 using MACS via a CNTN2 antibody and then grown on polymer coverslips in chamber slides (ibidi 80826) for 20–27 days. At the time of whole-cell patch clamping recordings, the polymer coverslips carrying RGCs were carefully cut out with a scalpel blade and then placed in a recording chamber under an upright microscope (Zeiss Examiner A1), containing artificial cerebrospinal fluid (aCSF) composed of (in mM): NaCl (140), MgCl$_2$ (1), KCl (5), CaCl$_2$ (2), Hepes (10), Glucose (10). Osmolarity and pH were adjusted to 300mOsm and 7.3 respectively. Whole-cell patch-clamp recordings in voltage and current-clamp mode were obtained at room temperature using an Optopatch amplifier (Cairn Research, UK) and acquired with WinWCP 5.2 freeware (John Dempster, SIPBS, University of Strathclyde, UK). Patch pipettes (3–4 MΩ when filled with corresponding solution) were pulled from borosilicate capillaries using a horizontal puller (Sutter P97, USA) and coated with dental wax to reduce the pipette capacitance. For current clamp recordings, patch pipettes were filled with a solution containing (in mM): K-gluconate (130), Na-gluconate (10), NaCl (4), Hepes (10), Phosphocreatin (10), MgATP (4), Na$_2$GTP (0.3). Osmolarity and pH were adjusted at 295mOsm and 7.3 respectively. Resting membrane potential was obtained from averaging membrane potential recorded for one minute in I=0 mode immediately after breaking the cell membrane. In current clamp mode, with the cell hyperpolarized to –70 mV, current steps of 100–500 pA were made to explore whether the cells were excitable. For voltage-clamp recordings, cells were held at clamped potential of –80 mV, and series resistance was monitored and compensated (>80%). Membrane potentials were corrected for the liquid junction potential, calculated at +5.4 mV (https://swharden.com/LJPcalc). Patch pipettes were filled with a solution containing (in mM): KCl (140), MgCl$_2$ (5), CaCl$_2$ (2.5), Hepes (10), MgATP (4), Na$_2$GTP (0.3). Osmolarity and pH were adjusted at 295mOsm and 7.3 respectively. A -P/4 subtraction protocol was used to isolate voltage-gated currents by removing linear leak current and capacitance artifacts. In all cases, voltage and current recordings were low pass filtered at 3 kHz and digitized at 10–20 kHz (Axon Digidata 1550b, Molecular Device, USA). Tetraethylammonium-chloride (TEA, Sigma-Aldrich, USA) and Tetrodotoxin-citrate (TTX, Fisher Scientific) stocks were diluted in aCSF to reach 20 mM and 1 µM respectively. All recordings were analyzed with WinWcp 5.2 and custom scripts and routines written in Python 3.9. Statistical tests were performed with corresponding function from *scipy.stats* package (v1.8).

## Statistical analysis

Statistical analysis in DEG identification was performed using the Seurat Package (v3.2.0) with default settings (*Stuart et al., 2019*). Adjusted p-values were shown.

## Acknowledgements

We greatly appreciate the Editors in *eLife* and anonymous Reviewers for enthusiasm and constructive critique. We are grateful to Dr. Roy Chuck for support, MRG Stem Cell Institute (supported by NYSTEM C029154) for service, Leslie Cummins and Frank P Macaluso at the Analytical Imaging Facility of AECOM for electron microscopy (supported by P30CA013330), David Reynolds at the Genomics Core of AECOM for the preparation of single-cell libraries, Dr. Kamran Khodakhah for technical support and critical readings, and grants from NEI, NIH (R01EY022645 and R21EY029806 to WL). Albert Lowe was supported by training grants T32GM007491 and C30292GG. Ludovic Spaeth was supported by a grant from NIH (R01NS105470 to Dr. Khodakhah).

## Additional information

### Competing interests

Wei Liu: A patent application is pending. (PCT/US2022/041843). The other authors declare that no competing interests exist.

## Funding

| Funder | Grant reference number | Author |
|---|---|---|
| National Eye Institute | R01EY022645 | Wei Liu |
| National Eye Institute | R21EY029806 | Wei Liu |
| National Institute of Neurological Disorders and Stroke | R01NS105470 | Ludovic Spaeth |
| National Institute of General Medical Sciences | T32GM007491 | Albert Lowe |
| The New York State Stem Cell Science program | C30292GG | Albert Lowe |

The funders had no role in study design, data collection and interpretation, or the decision to submit the work for publication.

## Author contributions

Wei Liu, Conceptualization, Resources, Data curation, Formal analysis, Supervision, Funding acquisition, Validation, Investigation, Visualization, Methodology, Writing – original draft, Project administration, Writing – review and editing; Rupendra Shrestha, Investigation, Writing – review and editing; Albert Lowe, Investigation; Xusheng Zhang, Software, Formal analysis, Investigation; Ludovic Spaeth, Resources, Data curation, Formal analysis, Validation, Investigation, Visualization, Methodology, Writing – original draft, Writing – review and editing

## Author ORCIDs

Wei Liu ⑪ https://orcid.org/0000-0003-4199-5406

Reviewer #1 (Public Review): https://doi.org/10.7554/eLife.87306.3.sa1
Reviewer #2 (Public Review): https://doi.org/10.7554/eLife.87306.3.sa2
Author Response https://doi.org/10.7554/eLife.87306.3.sa3

# Additional files

## Supplementary files

• Supplementary file 1. DEGs of cell clusters in the scRNA-seq dataset of CONCEPT organoids at day 24.

• Supplementary file 2. DEGs of cell clusters in the scRNA-seq dataset of human fetal retinas HGW9 (GSE138002).

• MDAR checklist

## Data availability

Single-cell RNA sequencing data of CONCEPT telencephalon-eye organoids at day 24 is available in Gene Expression Omnibus (http://www.ncbi.nlm.nih.gov/geo) using the accession number GSE191017. Scripts for analyzing single-cell RNA-seq datasets follow the tutorials of Seurat (https://satijalab.org/seurat/articles/get_started.html) and described in the section of single-cell RNA sequencing.

The following dataset was generated:

| Author(s) | Year | Dataset title | Dataset URL | Database and Identifier |
|---|---|---|---|---|
| Liu W, Shrestha R, Lowe A, Zhang X | 2023 | Self-formation of concentric zones of telencephalic and ocular tissues and directional retinal ganglion cell axons | https://www.ncbi.nlm.nih.gov/geo/query/acc.cgi?acc=GSE191017 | NCBI Gene Expression Omnibus, GSE191017 |

The following previously published dataset was used:

| Author(s) | Year | Dataset title | Dataset URL | Database and Identifier |
|---|---|---|---|---|
| Blackshaw S, Clark BS, Handa JT, Bremner R, Zack DJ | 2020 | scRNA-seq of the developing human retina | https://www.ncbi.nlm.nih.gov/geo/query/acc.cgi?acc=GSE138002 | NCBI Gene Expression Omnibus, GSE138002 |

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
