## [Editor Report · eLife assessment]

In this **important** study, the authors present a human telencephalon-eye organoid model that exhibits remarkable pathfinding and growth of retinal ganglion cell (RGC) axons. The identification of cell-surface markers for RGCs could have value for understanding the molecular mechanisms involved in RGC axon development and regeneration. The strength of evidence is **compelling** for future studies to investigate RGC neurite outgrowth and brain-eye connectivity in humans.

---

## [Referee Report · Reviewer #1 (Public Review)]

The authors set out to develop an organoid model of the junction between early telecephalic and ocular tissues to model RGC development and pathfinding in a human model. The authors have succeeded in developing a robust model of optic stalk(OS) and optic disc(OD) tissue with innervating retinal ganglion cells. The OS and OD have a robust pattern with distinct developmental and functional borders that allow for a distinct pathway for pathfinding RGC neurites.

Future work targeting RGC neurite outgrowth mechanisms will be exciting.

---

## [Referee Report · Reviewer #2 (Public Review)]

The authors compare their single-cell data of the self-forming brain-eye centroids with the published single-cell data from human fetal retinas and brain/optic organoids. This analysis further supports the similarity of their centroids with the human fetal retinal cell clusters, including the detection of the VSX2+/PAX2+ cells. The new findings further support the presented centroids' applicability for future studies on human RGC development and axon guidance mechanisms.

---

## [Author Response]

The following is the authors’ response to the original reviews.

Review 1Public ReviewThe authors set out to develop an organoid model of the junction between early telencephalic and ocular tissues to model RGC development and pathfinding in a human model. The authors have succeeded in developing a robust model of optic stalk(OS) and optic disc(OD) tissue with innervating retinal ganglion cells. The OS and OD have a robust pattern with distinct developmental and functional borders that allow for a distinct pathway for pathfinding RGC neurites.This study falls short on a thorough analysis of their single cell transcriptomics (scRNAseq). From the scRNAseq it is unclear the quality and quantity of the targeted cell types that exist in the model. A comparative analysis of the scRNAseq profiles of their cell-types with existing organoid protocols, to determine a technical improvement, or with fetal tissue, to determine fidelity to target cells, would greatly improve the description of this model and determine its utility. This is especially necessary for the RGCs developed in this protocol as they recommend this as an improved model to study RGCs.Future work targeting RGC neurite outgrowth mechanisms will be exciting.

We are grateful to Reviewer 1 for these constructive comments. We added plots for quality control in Figure 4-figure supplement 2 and quantification of cell clusters in Tab. 1. We compared the transcriptomes between CONCEPT organoids, Gabriel et al.’s brain/optic organoids (Gabriel et al., 2021; PMID: 34407456), and human fetal retinas HGW9 (Lu et al., 2020; PMID: 32386599), which strongly support our findings (Figs. 5, 6; see responses below for details). Besides FGFs/FGFR signaling, scRNA-seq identified additional candidate molecules that may provide axon guidance functions, and these candidate molecules are the focus of our future study.

Recommendations For The AuthorsThis study falls short on a thorough analysis of their single cell transcriptomics (scRNAseq).The scRNAseq figure needs to be better presented to allow for an adequate assessment of the model. As written the classification of the different clusters is hard to follow. A representative labeling of the suspected identity of the clusters in an infographic would aid the figure. Since it is hard to follow it is difficult to determine how well clusters correlate with designated cell types. PAX2 expression designating optic stalk seems to correlate well with the group 2 and the designation of the Optic disk, however PAX2 expression for the optic stalk is half in group 4 and half in group 9. what are group 4 and 9? It is also not clear how the thresholding for the given clusters was reached.

To present the scRNA-seq dataset in a clearer way, we added dotted red lines in Fig. 4C to delineate eye (mostly retinal), telencephalic, and mixed cell populations. In Tab. 1, we showed assigned cell types, counts, and percentage for each cluster.

PAX2+ VSX2- optic stalk cells were at edges of clusters 4, 8, 9 that had dorsal telencephalic identities. Clusters 4, 8, 9 were largely segregated along cell cycle phases (Fig. 4A, B, F), and these clusters differentially expressed gene markers SOX3, FGFR2, PRRX1, EDNRB, and FOXG1 (Figure 4-Figure supplement 4A-4D; Fig. 4C). In E14.5 mouse embryos, mouse orthologs of SOX3,FGFR2, PRRX1, and EDNRB were specifically expressed in dorsal telencephalon (Figure 4-figure supplement 5B-5E); Foxg1 was specifically expressed in both dorsal and ventral telencephalon. Therefore, clusters 4, 8, and 9 have dorsal telencephalic identities, and PAX2+ VSX2- optic stalk cells are at edges of these telencephalic clusters.

Thresholding of cell clusters were determined by cell clustering parameters, which is described in Materials and Methods: FindVariableFeatures (selection.method = "vst", nfeatures = 2000), ScaleData, RunPCA, ElbowPlot, FindNeighbors (dims = 1:17), FindClusters (resolution = 0.5), and RunUMAP(dims = 1:17).

The authors should make an attempt to calculate which different cell types are present and in what proportions. They should also discuss groups that are confounding. Since this is the first description of this technique it is critical to know how much of the model represents mature welldefined cells of interest.

We assigned cell types to clusters and calculated cell counts and proportions of each cluster (Tab.1). The only undetermined cell cluster was cluster 13, which was the smallest one. We described top DEGs of cluster 13 and discussed the cluster.

Concerning the focus on RGC isolation. It is interesting that CNTN2 can be used for an effective isolation however, there are many protocols for generating RGCs. Is CNTN2 expression unique to this protocol? If the authors claim that this protocol could be used for studying glaucoma, how does this protocol improve on the quality of RGCs compared to other protocols?

RGC-specific CNTN2 expression was not unique to CONCEPT organoids. We isolated RGCs via CNTN2 from both CONCEPT organoids and 3-D retinal organoids in suspension. Indeed, isolated RGCs shown in the manuscript were from 3-D retinal organoids (see Materials andMethods for details). Importantly, our single cell RNA sequencing analysis demonstrated that CNTN2 was also differentially expressed in early RGCs from human fetal retinas (Fig. 5L, 5M). Therefore, isolation of human early RGCs via CNTN2 should be applicable widely.

In CONCEPT organoids, RGC differentiation and directional axon growth were very efficient. Our study supports a model that FGFs from optic disc cells efficiently induce RGC differentiation and directional axon growth in adjacent retinal progenitor cells, as FGFR inhibitions drastically decreased the number of RGC somas and directional axon growth (Fig. 9). Therefore,CONCEPT organoids are useful in studying axon guidance cues in humans, which knowledge is much needed for axon regrowth from RGCs that are damaged in glaucoma. Notably, juvenile glaucoma gene CYP1B1 was found in assigned optic disc cells in both CONCEPT organoids and human fetal retinas (Fig. 4I, 5D), making CONCEPT organoids a testable model in studying the functions of CYP1B1 in human cells.

A comparative analysis of the scRNAseq profiles of their model with existing organoid protocols, to determine a technical improvement, or with fetal tissue, to determine fidelity to target cells, would greatly improve the description of this model and determine its utility.

In the revised manuscript, we compared the transcriptomes between CONCEPT organoids, Gabriel et al.’s brain/optic organoids (Gabriel et al., 2021; PMID: 34407456), and human fetal retinas HGW9 (Lu et al., 2020; PMID: 32386599). Gabriel et al. (2021) report “axon-like” projections in their “optic vesicle-containing brain organoids”. We found that PAX2+ VSX2+ optic disc, PAX2+ VSX2- optic stalk, FOXG1+ telencephalic, and VSX2+ neuroretinal cell clusters that were found in CONCEPT organoids did not exist in Gabriel et al.’s organoids (Figure 4-Figure supplement S1o), indicating striking differences between Gabriel et al.’s organoids and our CONCEPT telencephalon-eye organoids.

On the other hand, CONCEPT organoids and human fetal retinas HGW9 had similar expression signatures (Fig. 5). First, we identified a PAX2+ cell cluster in the human retinas HGW9. 64/113DEGs in the PAX2+ cluster from human fetal retinas HGW9 were also DEGs of cluster 2 (assigned PAX2+ optic disc cells) from CONCEPT organoids. Second, CNTN2 was also differentially expressed in early RGCs of human fetal retinas. Third, when cells in cluster 18 and retinal progenitor clusters from the HGW9 dataset were combined with cells in clusters 2, 4, 5, 7 from the CONCEPT dataset for Seurat anchor-based clustering, cells in cluster 18 from HGW9 (H18) were grouped with cluster 2 from CONCEPT organoids (C2, assigned optic disc; N), and these cells expressed both PAX2 and VSX2 (arrowheads in Fig. 5N-5R). A small portion of H18 cells were grouped with cluster 4 from CONCEPT organoids (C4, assigned optic stalk; N), and these cells expressed PAX2 but not VSX2 (arrows in Fig. 5N-5R). Fourth, CONCEPT organoids and human fetal retinas shared many enriched GO terms in DEGs of assigned optic disc cells (Fig. 6).

Collectively, transcriptomic comparisons support that our CONCEPT organoids are innovative and similar to human fetal retinas.

Not clear what reporting on Lens cells in Figure 3 adds to the focus of the manuscript. The figure seems out of place with the flow of the manuscript.

Lens cells were obvious in CONCEPT organoids. The presence of lens cells indicates that cysts have the developmental potential for both neural and non-neural anterior ectodermal cells. For a better flow, we added a transitional sentence at the beginning of the lens section.

Reviewer #2Public ReviewThe study by Liu et al. reports on the establishment and characterization of telencephalon eye structures that spontaneously form from human pluripotent stem cells. The reported structures are generated from embryonic cysts that self-form concentric zones (centroids) of telencephaliclike cells surrounded by ocular cell types. Interestingly, the cells in the outer zone of these concentric structures give rise to retinal ganglion cells (RGCs) based on the expression of several markers, and their neuronal morphology and electrophysiological activity. Single-cell analysis of these brain-eye centroids provides detailed transcriptomic information on the different cell types within them. The single-cell analysis led to the identification of a unique cellsurface marker (CNTN2) for the human ganglion cells. Use of this marker allowed the team to isolate the stem cell-derived RGCs.Overall, the manuscript describes a method for generating self-forming structures of brain-eye lineages that mimic some of the early patterning events, possibly including the guidance cues that direct axonal growth of the RGCs. There are previous reports on brain-eye organoids with optic nerve-like connectivity; thus, the novel aspect of this study is the self-formation capacity of the centroids, including neurons with some RGC features. Notably, the manuscript further reports on cell-surface markers and an approach to generating and isolating human RGCs.Recommendations For The AuthorsThe following significant issues, however, need to be addressed:The authors show RGC-like cells that grow axons toward the Pax2+ cells, suggesting that this is a model for RGC axon pathfinding. Is there support from transcriptomic data on the expression of guidance molecules? In addition, the authors need to characterize Pax2+ cells further. Do some give rise to astrocyte-like cells?

We assessed the expression of known axon guidance genes in CONCEPT organoids. FGF8 and FGF9 trigger axon outgrowth in motor neuron column explants (Shirasaki et al., 2006). In CONCEPT organoids, FGF8 and FGF9 were differentially expressed in assigned optic disc cells; FGFR inhibition drastically decreased the number of RGC soma and directional axon growth (Fig. 9). In addition, SEMA5a and EFNB1 were expressed in both assigned optic disc and stalk cells, EFNB2 was highly expressed in assigned optic disc cells, and NTN1 was mostly expressed in assigned optic cells (Figure 4-figure supplement 9).

We compared the transcriptomes between CONCEPT organoids, Gabriel et al.’s brain/optic organoids (Gabriel et al., 2021; PMID: 34407456), and human fetal retinas HGW9 (Lu et al., 2020; PMID: 32386599). Gabriel et al. (2021) report “axon-like” projections in their “optic vesicle-containing brain organoids”. We found that PAX2+ optic disc, PAX2+ optic stalk, FOXG1+ telencephalic, and VSX2+ neuroretinal cell clusters that were found in CONCEPT organoids did not exist in Gabriel et al.’s organoids (Figure 4-figure supplement 10), indicating striking differences between Gabriel et al.’s organoids and our CONCEPT telencephalon-eye organoids.

To authenticate PAX2+ cells in CONCEPT organoids, we analyzed a single-cell RNA-seq dataset of human fetal retinas HGW9 and identified a similar PAX2+ cell population, cluster 18 (Fig. 5). Expression signatures of PAX2+ cells between CONCEPT organoids and human fetal retinas HGW9 were similar. Notably, cluster 18 differentially expressed PAX2, COL9A3, CYP1B1, SEMA5A, and FGF9 (Fig. 5B-5F), which were top DEGs of cluster 2 in CONCEPT organoids (Fig. 4F, 4G, 4I, 4K; SEMA5A was shown in Figure 4-figure supplement 9A). Overall, 64/113 DEGs of cluster 18 in human fetal retinas HGW9 were also DEGs of cluster 2 in CONCEPT organoids. In both HGW9 and CONCEPT organoids, expression of OLIG2, CD44, and GFAP was undetectable (Figure 4-figure supplement 11), indicating that astrocytes had not been generated yet at these stages.

When cells in cluster 18 and retinal progenitor clusters from the HGW9 dataset were combined with cells in clusters 2, 4, 5, 7 from the CONCEPT dataset for Seurat anchor-based clustering, cells in cluster 18 from HGW9 (H18) were grouped with cluster 2 from CONCEPT organoids (C2, assigned optic disc; N), and these cells expressed both PAX2 and VSX2 (arrowheads inFig. 5N-5R). A small portion of H18 cells were grouped with cluster 4 from CONCEPT organoids (C4, assigned optic stalk; N), and these cells expressed PAX2 but not VSX2 (arrows in Fig. 5N5R).

We then compared functional annotations of DEGs (top 200 genes) of cluster 2 in CONCEPT organoids and DEGs (113 genes) of cluster 18 in human fetal retinas HGW9. Top GO terms in GO:MF, GO:CC, and GO:BP are shown (Fig. 6). For DEGs of cluster 2 in CONCEPT organoids, top enriched GO terms in GO:MF, GO:CC, and GO:BP were extracellular matrix structural constituent, collagen-containing extracellular matrix, and system development, respectively. Additional interesting GO:BP terms included axon development, astrocyte development, eye development, response to growth factor, cell adhesion, cell motility, neuron projection development, glial cell differentiation, and signal transduction. For DEGs of cluster 18 in human fetal retinas HGW9, top enriched GO terms in GO:MF, GO:CC, and GO:BP were cell adhesion molecule binding, extracellular space, and developmental process, respectively. Many GO terms were enriched in both samples, further indicating transcriptomic similarities in PAX2+ optic disc cells between CONCEPT organoids and human fetal retinas. Notably, GO terms astrocyte differentiation, neuron projection development, and glial cell differentiation were enriched in the DEGs of assigned optic disc cells for both CONCEPT organoids and human fetal retinas, consistent with expectations.

The Vsx2+Pax2+ population is not typically detected in vivo in the developing mouse eye. The authors claim that they detected them in vivo, but the data supporting this statement are lacking.

We demonstrate that assigned optic disc cells expressed both VSX2 and PAX2, and this statement is trued for CONCEPT organoids and human fetal retinas HGW9 (Fig. 5N-5R).

Do the RGCs express subtype-specific markers? Do they detect markers of other retinal neurons typically born early in development-cones, amacrine cells, horizontal cells? The authors need to compare the transcriptome of different clusters to the published datasets from human and mouse retinae.

The stage of CONCEPT organoids for scRNA-seq was at an early stage. In this dataset, subtypes of RGCs were undetectable. Isolated RGCs via CNTN2 were at more advanced stages. Distinct expression of POU4F2, ISL1, RBPMS, and SNCG indicate multiple subtypes of RGCs (Fig. 7L-7P).

We did find other early retinal neurons in the scRNA-seq dataset: photoreceptor cells, amacrine/horizontal cells in CONCEPT organoids (Fig. 4U-4X), and these cells were also in cluster 11 in which RGCs were found.

We performed transcriptomic comparisons between CONCEPT organoids, brain/optic organoids, and human fetal retinas. We found that PAX2+ optic disc, PAX2+ optic stalk, FOXG1+ telencephalic, and VSX2+ neuroretinal cell clusters that were found in CONCEPT organoids did not exist in Gabriel et al.’s organoids, indicating striking differences between Gabriel et al.’s organoids and our CONCEPT telencephalon-eye organoids (Figure 4-figure supplement 10). On the other hand, we found that expression signatures of CONCEPT organoids and human fetal retinas are similar (Figs. 5, 6).

Fig. 3: where are the "lens like" cells located? The structures in panels B and D look very different. Are these lens-cells toward the periphery or scattered throughout?

Lens cells were dispersed in the zone in which neural retinal cells are located, which is shown in a low-magnification image (Fig. 3K). Panel B and D in Figure 3 were at different stages. At early stages, lens clusters were small (Fig. 3B). At later stages, lens clusters became bigger (Fig. 3D).

Fig. 3K and L, TEM images: how do the authors know that these are lens cells?

Western blot of these transparent cell clusters demonstrated that they were lens cells (Fig. 3L).

Fig. 5: The authors claim that a reduced number of Pax2+ cells is associated with entry of the axons. It is not clear if this is just due to physical barriers or to active axon guidance.

We believe that Reviewer 2 referred to the gap region of PAX2 expression in Fig. 7A, 7F. RGC axons grew toward and along adjacent PAX2+ VSX2+ cells. Since PAX2+ VSX2+ cells grossly formed a circular shape, RGC axons followed this circular shape. In a gap region of PAX2 expression, RGC axons exited the circle. The association of RGC axon growth with PAX2+ VSX2+ cells was very robust. Besides PAX2+ cell populations, we did not find any other cell populations that directed RGC axon growth.

Fig. 5K: The authors refer to ALDH1A3 expression in the optic disk, but the presented section does not include the optic disk. In addition, ALDH1A3 is expressed in other regions of the developing retina (Fig. 5K, ref 71).

We are sorry we did not make it clear. We referred to Li et al.’s (2000) paper (Mech Dev 95, 283-289) for Aldh1a3 expression in the optic stalk. Figure 7K was used to shown Aldh1a3 expression in peripheral retinas on sections.

Line 263, Reference 68: The authors claim that col13A1 is specific to the human optic disk. However, col13A1 is expressed in many additional eye lineages (PMID: 10865988).

We are sorry we did not make it clear. We meant that Col13A1 is prominently expressed in the optic disc, which is clearly shown in the referred paper (Figure 3D in the paper PMID: 10865988).

The authors show that inhibiting FgfR results in fewer RGCs and loss of directed axonal growth. The number of cells is drastically reduced; thus, the relevance of the finding directly to axon guidance is not resolved.

FGFR inhibitions drastically the number of RGC somas (Fig. 9F-9K). Additionally, remaining RGCs nearly did not grow directional axons (arrowheads in Fig. 9K), and a few remaining axons wandered around (arrow in Fig. 9K), indicating the role of FGF/FGFR signaling in early RGC differentiation and directional axon growth.

Fig. 1H and J: Vsx2 is outside the centroid in panels H and I, but inside the centroid in panels J and K. It is not clear what part of the centroid is shown. This needs to be clarified by adding a scheme.

We are sorry we did not make it clear. We added separate-channel images showing VSX2 andPAX6 expression (Figure 1-figure supplements 1, 2) and a new diagram (left panel in Fig. 1B). Overall, FOXG1, VSX2, and PAX6 expression at days 15-17 formed three concentric zones spanning from the center to the periphery. At days 22-26, VSX2 expression expanded peripherally, largely overlapping PAX6 expression (Figure 1-figure supplements 1, 2).

Pax6 should be in all cells, also on day 17. Show the separate channels, including DAPI.

We added separate-channel images (Figure 1-figure supplements 1, 2). In cysts, PAX6 was expressed in all cells. After cysts attached to the culture surface and grew as colonies, distinct levels of PAX6 expression emerged in concentric zones. At days 17 and 26, PAX6 expression at the central zone (which cells expressed FOXG1) became lower, which is obvious in separate-channel images (Figure 1-figure supplements 1, 2). Consistently, PAX6 expression was low in FOXG1+ telencephalic cells in the scRNA-seq (Fig. 4C, 4D).

Lines 27-30: this is a long and complex sentence which needs to be clarified.

We broke it into a few sentences to make it clearer.

Line 43: fix "Retina" to "Retinal"

We fixed it.

Lines 376-377: repeated "mechanisms of".

We fixed it.